# TiMi: Empowering Time Series Transformers with Multimodal Mixture of Experts

## Abstract

Multimodal time series forecasting has garnered significant attention for its potential to provide more robust and accurate predictions than traditional single-modality models by leveraging rich information inherent in other modalities. However, due to fundamental challenges in modality alignment, existing methods often struggle to effectively incorporate multimodal data into predictions, particularly textual information that has a causal influence on time series fluctuations, such as emergency reports and policy announcements. In this paper, we reflect on the role of textual information in numerical forecasting and propose **Ti**me series transformers with Multimodal **Mi**xture-of-Experts, **TiMi**, to unleash the causal reasoning capabilities of LLMs. Concretely, TiMi utilizes language models to generate inferences on future developments, which then serve as guidance for time series forecasting. To seamlessly integrate both exogenous factors and time series into predictions, we introduce a Multimodal Mixture-of-Experts (MMoE) module as a lightweight plug-in to empower Transformer-based time series models for multimodal forecasting, eliminating the need for explicit representation-level alignment. Experimentally, our proposed TiMi demonstrates consistent state-of-the-art performance on sixteen real-world multimodal forecasting benchmarks, outperforming advanced unimodal and multimodal baselines while offering strong adaptability and interpretability.

## 1 Introduction

The rapid development of deep learning has revolutionized time series analysis, driving significant progress across various real-world applications, such as energy management (Weron, 2014), transportation (Cai et al., 2020), healthcare (Kaushik et al., 2020), and meteorology (Wu et al., 2023b). However, a key limitation persists that most existing methods remain largely confined to numerical time series, overlooking the rich, complementary information available in other modalities.

Since time series data represent a partial observation of a complex system (Wang et al., 2024), solely relying on historical series is usually insufficient for accurate prediction. Incorporating exogenous factors from other modalities is therefore essential, as they describe the dynamic behaviors of the system and enhance forecasting performance. For example, in e-commerce sales forecasting (Li et al., 2024), sales fluctuations are strongly influenced by promotional campaigns, whose details are often embedded in textual data such as product descriptions, advertisements, or marketing announcements. Recent advances in Large Language Models (LLMs) (Radford et al., 2019; 2021; Yang et al., 2025), which excel at extracting and reasoning over textual information (Yin et al., 2024), present new opportunities for time series analysis to surpass the limitations of traditional unimodal approaches.

While the integration of multimodal data holds immense promise, current research has yet to fully leverage the full potential of textual information for time series forecasting. Following methodologies from well-established vision-language models (Li et al., 2021; Liu et al., 2023), existing multimodal time series forecasters primarily focus on cross-modality alignment (Jia et al., 2024) between image and text. However, such approaches overlook a key distinction: textual and visual modalities are inherently semantically aligned, whereas textual information related to numerical observations is typically based on future expectations. Some studies have explored aligning metadata text with time series, as the two can offer mutually descriptive perspectives. But this line of work remains limited, as metadata only reflects the internal attributes of sequences and fails to introduce external causal factors of fluctuations in time series.

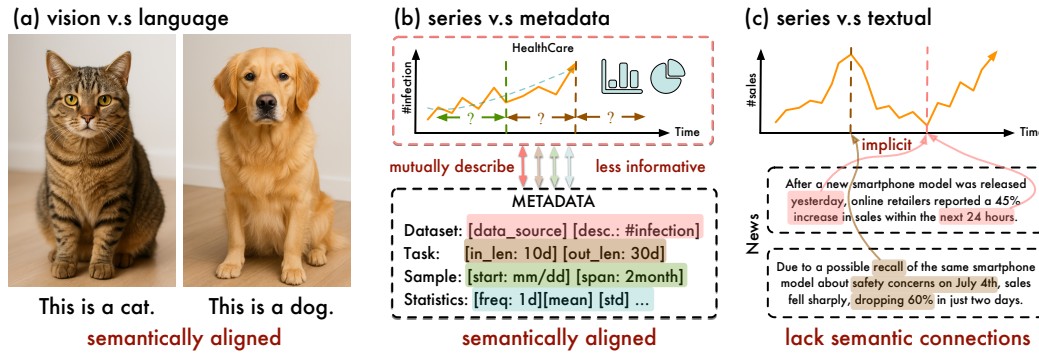

Figure 1: Two types of multimodal time series data exist in real-world forecasting scenarios. Compared to series-metadata data, which is semantically aligned like vision-language data, series-textual data lacks this direct alignment but often offers richer and more informative insights for prediction.

On the other hand, external textual information in real-world scenarios, such as news or policy announcements that explain fluctuations, lacks direct semantic correspondence, posing greater challenges for multimodal time series forecasting. Simply applying alignment techniques, such as modality fusion (Jin et al., 2023), can lead to suboptimal performance and limited interpretability (Tan et al., 2024). A key reason is that textual content itself contains irrelevant information that contributes little to forecasting. Therefore, the implicit knowledge embedded within textual data, such as the causal factors and prior understanding of future events, needs to be mined to guide the model toward more accurate predictions.

In this paper, we introduce a paradigm-shifting approach in multimodal time series forecasting, **Ti**me series Transformers with multimodal **Mi**xture of experts (**TiMi**). Specifically, we utilize the reasoning capabilities of LLMs to extract structured causal knowledge of future trends from textual content. To effectively integrate extracted information as a form of future guidance into more context-aware predictions, we introduce Multimodal Mixture-of-Experts (MMoE) as a lightweight plug-in module tailored for Transformer-based time series forecasters. Leveraging the selective routing mechanism of MoE, a text-informed MoE (TMoE) is introduced to seamlessly channel textual information into the modeling process, enabling adaptive representation learning guided by causal knowledge while avoiding vague feature-level fusion. Meanwhile, to capture the long-term trends within series that are equally vital for forecasting, we introduce a series-aware MoE (SMoE) that attends to historical series and provides complementary guidance together with TMoE for time series prediction.

Our proposed framework effectively bridges distinct modalities, offering a robust solution for multimodal time series forecasting without introducing the limitations of traditional multimodal fusion techniques. Our contributions are summarized as follows:

- We revisit fundamental challenges in multimodal time series forecasting and propose TiMi, a novel framework that leverages the causal reasoning capabilities of LLMs to extract structured causal knowledge of future trends from textual content.

- We introduce the Multimodal Mixture-of-Experts (MMoE) as a lightweight plug-in module into Transformer-based time series forecasters, which seamlessly integrate structured extracted knowledge into context-aware predictions instead of vague feature-level fusion.

- Extensive experiments on sixteen real-world forecasting benchmarks validate the effectiveness of TiMi, which consistently achieves state-of-the-art performance and outperforms advanced unimodal and multimodal baseline with adaptability and interpretability.

## 2 RELATED WORK

### 2.1 TRANSFORMER-BASED TIME SERIES FORECASTING

In recent years, Transformer architecture (Vaswani et al., 2017) has demonstrated impressive achievements in areas such as natural language processing and computer vision. Benefiting from the attention

mechanism, these models possess a powerful capacity for relation modeling in sequential data. Given the sequential nature of time series data, Transformer has also achieved substantial progress in time series analysis. Early efforts, using vanilla Transformer architecture, were limited by its high computational complexity and lack of inductive bias for sequential data. Consequently, a substantial body of work has been introduced to enhance the architecture's effectiveness in time series analysis.

Early works, represented by LogSparse (Li et al., 2019) and Informer (Zhou et al., 2021), renovate the self-attention mechanism to overcome the computational efficiency bottleneck. Furthermore, Autoformer (Wu et al., 2021) proposes an Auto-Correlation mechanism that discovers dependencies and aggregates information at the series level. Similarly, recent studies capture series-level temporal dependencies by dividing time series into multiple patches and modeling the interdependencies between different patches. As an extreme case of series-level dependencies, iTransformer (Liu et al., 2024c) revisits the mechanism by treating the entire series as a single token and thereby learn the multivariate correlations. More recently, TimeXer (Wang et al., 2024) introduces a learnable global token that interacts with tokens of different granularities, enabling the Transformer to simultaneously model temporal dependencies within endogenous series and correlations between variables. This meticulous design underscores the capability of Transformer architecture to handle heterogeneous time series data and highlights a promising direction for modeling multimodal time series data.

## 2.2 MULTIMODAL TIME SERIES FORECASTING

With the rapid advancement of large language models (LLMs), multimodal time series forecasting has gained increasing attention, particularly the integration of textual content. Based on the differences in modality fusion, existing research can be divided into two categories.

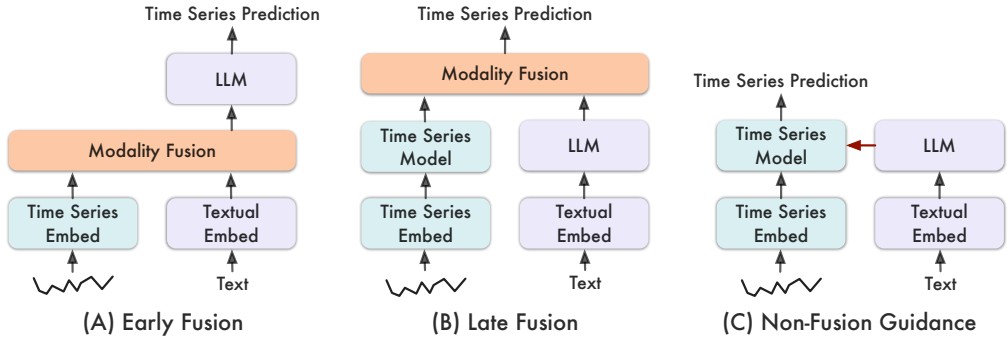

Figure 2: Three categories of multimodal time series forecasting models distinguished through fusion.

One line of work focuses on *Early Fusion*, which aligns time series data into a textual hidden state and takes LLMs as the backbone to leverage the powerful modeling capabilities of LLMs. For instance, Time-LLM (Jin et al., 2023) reprograms time series data into text prototypes while keeping the LLM backbone model intact. UniTime (Liu et al., 2024b) directly concatenates temporal and textual features through embedding and inputs them into a pre-trained large model to adapt to multimodal data in multiple fields. Similarly, AutoTimes (Liu et al., 2024d) introduces an innovative tokenization method for time series data, projecting time series into the embedding space of language tokens.

Another type of work, termed *Late Fusion*, processes time series data with specialized time series models and textual content with large language models, separately generating distinct features for each modality. Time-MMD (Liu et al., 2024a) exemplifies this approach by employing a multi-layer perceptron (MLP) after the LLM to map textual features into time series prediction. Moreover, IMM-TSF (Chang et al., 2025) extends this idea and proposes Timestamp-to-Text and Multimodality Fusion modules to align time series, timestamp, and textual features for improved prediction performance.

Notably, multimodal data in real-world scenarios often face semantic misalignment between textual and numerical modalities, posing significant challenges for previous approaches to achieve explicit alignment. In this paper, we introduce a *Non-Fusion Guidance* approach for multimodal time series forecasting, as illustrated in Figure 2, which leverages textual information to guide the time series backbone model in making predictions.

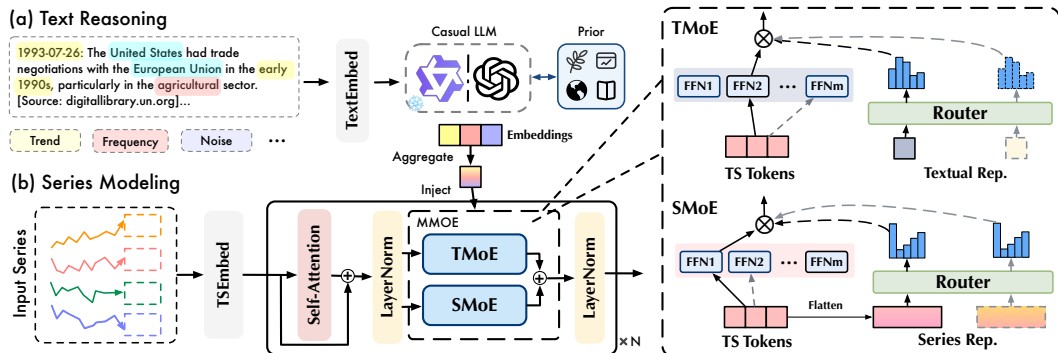

Figure 3: Overall design of TiMi, a time series-centric Transformer model. Textual content is encoded by LLMs to infer future trends of predictions. The MMoE module integrates a global view of both historical input and causal knowledge derived from text, thereby enhancing the time series modeling.

## 3 TIMI

### 3.1 PROBLEM FORMULATION

In canonical time series forecasting, the objective is to predict the future values of a time series over the future time steps solely based on the historical observation $\mathbf{x}_{1:L} = \{x_1, x_2, ..., x_L\} \in \mathbb{R}^{L \times C}$, where $L$ represents the length of the lookback window, and $C$ denotes the number of variable. By contrast, multimodal time series forecasting extends this task by incorporating data from other modalities, such as exogenous textual content denoted as $\mathcal{T}$. This contextual information provides critical supplementary signals for prediction. Therefore, given a forecasting model $\mathcal{F}$, parameterized by $\theta$, the goal of the multimodal forecasting task is to forecast the time series for the next $S$ time steps by leveraging both historical observations and exogenous textual information, which can be formulated as: $\hat{\mathbf{x}}_{L+1:L+S} = \mathcal{F}_\theta(\mathbf{x}_{1:L}, \mathcal{T})$.

### 3.2 MULTIMODAL EMBEDDING

As depicted in Figure 3, our proposed TiMi is a Transformer-based time series-centric multimodal forecasting model. Technically, we treat time series as the primary modality and design a Mixture of Experts (MoE) module embedded in the Transformer backbone to guide time series forecasting by integrating causal knowledge from both textual and series perspectives.

**Text Encoding** In time series forecasting, textual data often serves as important exogenous factors, providing complementary context and valuable insights about potential future developments. Trained on diverse and large-scale corpora, large language models (LLMs) demonstrate exceptional capabilities in reasoning, contextual understanding, and extrapolation, which present a promising approach for extracting such insights. TiMi employs a frozen large language model to process exogenous textual content and generate prediction embeddings of multifaceted temporal characteristics, such as trends, periodicity, and fluctuations. The outputs of these embeddings are then aggregated into a textual token through average pooling with rich causal knowledge, enabling further interaction with temporal tokens. Given $\bar{\mathcal{H}}$ as the hidden representation of textual information, the overall process can be formalized as:

$$\bar{\mathcal{H}} = \text{AvgPool}(\text{LLM}(\mathcal{T})) \tag{1}$$

**Time Series Embedding** As a time series-centric model, TiMi learns and exploits patch-wise representations of historical time series to capture future temporal variations more precisely. Concretely, the series is split into $N$ overlapping patches of length $P$, where $N = \lfloor \frac{L}{P} \rfloor$ represents the number of patches. Each patch is then projected into a temporal token $\mathbf{h}$ via a trainable linear projector $\text{PatchEmbed}(\cdot) : \mathbb{R}^P \to \mathbb{R}^D$, which is formally stated as:

$$\begin{aligned} \{\mathbf{s}_1, \mathbf{s}_2, ..., \mathbf{s}_N\} &= \text{Patchify}(\mathbf{x}), \\ \{\mathbf{h}_i\}_{i=1}^N &= \text{PatchEmbed}(\{\mathbf{s}_i\}_{i=1}^N), \end{aligned} \tag{2}$$

## 3.3 MULTIMODAL MIXTURE OF EXPERTS

The exogenous textual content presents the future expectation of time series data, whereas, without considering the current time series context, the Mixture-of-Experts may not closely connect causal information with the series. Building upon this, we propose a novel Multimodal Mixture-of-Experts (MMoE) module that leverages complementary information from both time series and textual data to enhance forecasting performance. Specifically, MMoE comprises a text-informed Mixture-of-Experts (TMoE), routed based on the textual representation of future-relevant insights, and a series-aware Mixture-of-Experts (SMoE), guided by global representations of historical temporal dynamics. The two perspectives distinctly provide guiding suggestions according to the context of different modalities, and the representations for forecasting are constructed through a fusion of the Multimodal Mixture-of-Experts. The overall process can be formalized as follows:

$$\hat{\mathbf{h}}^l = \text{LayerNorm}(\mathbf{h}^{l-1} + \text{Self-Attention}(\mathbf{h}^{l-1})),$$
$$\mathbf{h}^l = \text{LayerNorm}(\hat{\mathbf{h}}^l + \text{MMoE}(\hat{\mathbf{h}}^l, \bar{\mathcal{H}})). \tag{3}$$

**Text-Informed Mixture of Experts**  In multimodal time series forecasting, exogenous textual information provides valuable contextual insights that enhance the prediction of time series data. Unlike previous approaches that focused primarily on cross-modal alignment, TiMi leverages the causal reasoning capabilities of large language models (LLMs) to infer future temporal variations and thereby inform its predictions. Specifically, the textual embeddings are used as inputs to a gating network of the TMoE module, which dynamically selects a sparse combination of experts shared across all temporal tokens. This design enables the model to effectively distill knowledge of the future from textual data and incorporate it into time series prediction, facilitating a richer contextual understanding and more accurate forecasting.

$$s_{i,t} = \text{Softmax}_i(\mathbf{W}_t \bar{\mathcal{H}}),$$
$$\text{TMoE}(\mathbf{h}, \bar{\mathcal{H}}) = \sum_{i \in \tau_t} s_{i,t} \, \text{FFN}_i(\mathbf{h}), \tag{4}$$

where $\mathbf{W}_t^l \in \mathbb{R}^{M \times \bar{D}}$ denotes the trainable parameters, where $M$ is the number of experts and $\bar{D}$ is the hidden dimension of textual data, $\tau_t$ is the set of selected top-$k$ indices of experts. By dynamically selecting experts for specific functions, TMoE leverages future trends extracted from textual content to provide targeted guidance for time series prediction.

**Series-Aware Mixture of Experts**  Beyond the incorporation of textual information, we further introduce an SMoE module tailored for temporal data to provide a global view of the series. Specifically, patch-level temporal tokens are concatenated into a series-level representation to capture the overarching temporal patterns of the series. This global representation is then used to route a shared set of experts across all patch tokens, enabling the model to develop a holistic understanding of the series, in addition to learning patch-wise temporal dependencies.

$$s_{i,s} = \text{Softmax}_i(\mathbf{W}_s[\mathbf{h}_1, \cdots, \mathbf{h}_N]),$$
$$\text{SMoE}(\mathbf{h}) = \sum_{i \in \tau_s} s_{i,s} \, \text{FFN}_i(\mathbf{h}). \tag{5}$$

Here, $\mathbf{h}_i$ is the $i$-th patch token and $[\cdot]$ denotes the concatenation of all patch tokens along the hidden dimension. By concatenating all patch token together, SMoE can obtain the global series representation, $\mathbf{W}_s^l \in \mathbb{R}^{M \times (N \times D)}$. $\tau_s$ are the set of selected top-k indices of experts.

By combining a multifaceted view of the data through the multimodal mixture-of-experts module, TiMi enables more informed predictions. The overall process can be summarized as follows:

$$\text{MMoE}(\mathbf{h}, \bar{\mathcal{H}})) = \sum_{i \in \tau_x} s_{i,x} \, \text{FFN}_i(\mathbf{h}) + \sum_{i \in \tau_s} s_{i,s} \, \text{FFN}_i(\mathbf{h}). \tag{6}$$

## 4 EXPERIMENTS

We extensively validate the effectiveness of our proposed TiMi on a wide range of real-world multimodal time series forecasting benchmarks spanning sixteen datasets from diverse domains.

**Datasets** For a comprehensive evaluation, we employ the Time-MMD dataset (Liu et al., 2024a), a multi-domain, multimodal time series dataset encompassing nine diverse real-world scenarios, including Agriculture, Climate, Economy, Energy, Health(US), Security, SocialGood, and Traffic. Each dataset comprises numerical data with relevant textual data, which is highly correlated to the prediction of time series.

**Baselines** We include ten state-of-the-art deep forecasting models as our baselines, including multimodal time series models: Time-MMD (Liu et al., 2024a), AutoTimes (Liu et al., 2024d), Time-LLM (Jin et al., 2023), GPT4TS (Zhou et al., 2023), Transformer-based time series models: TimeXer (Wang et al., 2024), iTransformer (Liu et al., 2024c), PatchTST (Nie et al., 2022), Autoformer (Wu et al., 2021), CNN-based model: TimesNet (Wu et al., 2023a), and Linear-based model: DLinear (Zeng et al., 2023).

**Implementation Details** To give a clear comparison between TiMi with baselines, we follow the forecasting setup in Time-MMD (Liu et al., 2024a). Specifically, for monthly-sampled datasets such as Agriculture, Climate, Economy, Security, SocialGood, and Traffic, the lookback length of time series is fixed at $L = 8$ and the prediction horizons are $H \in \{6, 8, 10, 12\}$. For weekly-sampled datasets such as Energy and Health(US), the input length is set to $L = 24$, while the prediction horizons are $H \in \{12, 24, 36, 48\}$. For the daily-sampled Environment dataset, the input length $L = 96$ with the prediction length $H$ varies in $\{48, 96, 192, 336\}$. For all multimodal baselines, we use GPT-2 (Radford et al., 2019) as the language model backbone, and adopt PatchTST as the time series backbone for TiMi and TimeMMD to ensure fair comparison.

### 4.1 MAIN RESULTS

Experimental results are presented in Table 1, where TiMi consistently achieves superior forecasting performance across most datasets. Notably, the multimodal baselines struggle to exhibit significant advantages over advanced unimodal time series forecasters. For instance, AutoTimes, which aligns numerical data into a textual latent space, fails to leverage external textual information to enhance its predictions and therefore shows underwhelming performance across all datasets. Moreover, even with the inclusion of external text, Time-MMD yields only marginal gains compared to its backbone time series model PatchTST. In contrast, our proposed TiMi, which also employs PatchTST for time series modeling, capitalizes on the reasoning capabilities of large language models (LLMs) to derive more profound insights into future temporal dynamics, demonstrating substantial performance gains over both unimodal and multimodal forecasting models. Full results are shown in Appendix G.4.

Table 1: Multimodal time series forecasting results on Time-MMD datasets with the best in **red** and the second underlined. Based on the varying sampling frequencies of each dataset, there are three fixed input lengths of 8, 24, and 96, respectively corresponding to three sets of prediction lengths $\{6, 8, 10, 12\}$, $\{12, 24, 36, 48\}$, and $\{48, 96, 192, 336\}$. The table reports the average results across all prediction lengths.

| Models | TiMi (Ours) | | Time-MMD (2024a) | | AutoTimes (2024d) | | Time-LLM (2023) | | GPT4TS (2023) | | TimeXer (2024) | | iTransformer (2024c) | | PatchTST (2022) | | TimesNet (2023a) | | DLinear (2023) | | Autoformer (2021) | |
|---|---|---|---|---|---|---|---|---|---|---|---|---|---|---|---|---|---|---|---|---|---|---|
| Metric | MSE | MAE | MSE | MAE | MSE | MAE | MSE | MAE | MSE | MAE | MSE | MAE | MSE | MAE | MSE | MAE | MSE | MAE | MSE | MAE | MSE | MAE |
| Agriculture | **0.193** | **0.299** | 0.213 | 0.301 | 1.032 | 0.716 | 0.262 | 0.318 | 0.224 | 0.306 | 0.230 | 0.306 | 0.227 | 0.306 | 0.242 | 0.312 | 0.217 | 0.316 | 0.511 | 0.514 | 0.282 | 0.350 |
| Climate | **0.872** | **0.737** | 0.968 | 0.783 | 0.944 | 0.758 | 0.915 | 0.750 | 0.928 | 0.756 | 0.950 | 0.767 | 0.970 | 0.779 | 0.977 | 0.779 | 0.930 | 0.767 | 0.915 | 0.750 | 0.943 | 0.763 |
| Economy | **0.012** | **0.086** | 0.022 | 0.123 | 0.736 | 0.729 | 0.018 | 0.110 | 0.014 | 0.096 | 0.013 | 0.091 | **0.012** | 0.087 | 0.013 | 0.093 | 0.014 | 0.096 | 0.086 | 0.237 | 0.061 | 0.194 |
| Energy | **0.229** | **0.344** | 0.254 | 0.365 | 0.335 | 0.432 | 0.266 | 0.380 | 0.280 | 0.397 | 0.247 | 0.361 | 0.253 | 0.364 | 0.288 | 0.392 | 0.271 | 0.383 | 0.310 | 0.418 | 0.306 | 0.418 |
| Environment | 0.408 | 0.462 | 0.471 | 0.515 | 0.441 | 0.490 | 0.414 | 0.464 | 0.423 | 0.469 | **0.401** | 0.463 | 0.419 | 0.465 | 0.417 | 0.469 | **0.401** | **0.458** | 0.431 | 0.507 | 0.441 | 0.490 |
| Health(US) | **1.194** | **0.745** | 1.349 | 0.821 | 1.387 | 0.798 | 1.241 | 0.754 | 1.241 | 0.761 | 1.279 | 0.781 | 1.251 | 0.756 | 1.252 | 0.759 | 1.301 | 0.784 | 1.390 | 0.789 | 1.313 | 0.804 |
| Security | **71.519** | **3.896** | 78.273 | 4.358 | 99.263 | 5.621 | 74.369 | 4.028 | 86.022 | 4.683 | 78.924 | 4.305 | 79.724 | 4.365 | 84.305 | 4.560 | 92.013 | 4.885 | 75.383 | 4.116 | 83.722 | 4.379 |
| SocialGood | **0.886** | **0.449** | 1.103 | 0.497 | 0.992 | 0.594 | 1.004 | 0.481 | 1.029 | 0.484 | 1.302 | 0.514 | 1.283 | 0.515 | 1.293 | 0.523 | 1.374 | 0.532 | 1.043 | 0.574 | 1.131 | 0.559 |
| Traffic | **0.154** | 0.218 | 0.277 | 0.317 | 0.187 | 0.288 | 0.174 | 0.228 | 0.192 | 0.244 | 0.176 | 0.220 | 0.179 | 0.220 | 0.173 | 0.219 | 0.168 | **0.215** | 0.189 | 0.306 | 0.165 | 0.230 |

## 4.2 TiMi GENERALITY

**Extending to irregular multimodal data** Irregular multimodal time series forecasting is a challenging yet prevalent problem in real-world scenarios, where data from different modalities are often temporally misaligned. TiMi possesses a strong potential to handle such irregular data since it models the time series and textual data separately, thus circumventing the need for perfect temporal alignment. Therefore, we employ a real-world irregular multimodal time series dataset (Time-IMM) (Chang et al., 2025) to validate its generality. Technologically, we adhere to Time-IMM's training and evaluation framework for fair comparison. We include advanced multimodel forecasting models, including IMM-TSF (Chang et al., 2025) and Time-MMD (Liu et al., 2024a), as baselines. Each baseline is implemented with PatchTST as a time series backbone.

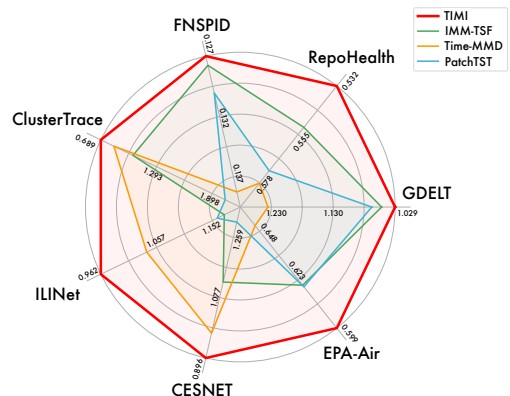

Figure 4: Irregular multimodal time series forecasting results on Time-IMM datasets. The metric for comparison is MSE. We compared four methods using PatchTST as the temporal backbone, among which the baseline results of IMM-TSF and PatchTST were derived from Time-IMM (Chang et al., 2025).

As illustrated in the Figure 4, TiMi consistently outperforms the baselines in terms of mean squared error (MSE) across all seven irregular datasets, with a substantial performance improvement on datasets such as ClusterTrace and CESNET. By employing LLMs for language modeling, TiMi is capable of handling textual data regardless of the temporal misalignment. Specifically, compared to the time series backbone PatchTST, TiMi reduces the average mean squared error by 29.57%, significantly outperforming Time-MMD (11.26%) and Time-IMM (16.46%).

**Adaptive Study** With the proposed MMoE module to bridge textual information into time series modeling, TiMi effectively integrates future insights into time series modeling, thereby enhancing prediction accuracy. It is worth noticing that the proposed MMoE is a plug-in model that can be applied to various Transformer-based forecasters.

*(i) Varying Transformer backbone.* To validate the generality of MMoE, we apply it to several Transformer-based time-series models, including TimeXer (Wang et al., 2024), PatchTST (Nie et al., 2022), and Autoformer (Wu et al., 2021). Specifically, we replace the standard FFN layers in both the encoder and decoder with the proposed MMoE module. The experimental results are shown in the Table 2. We can find that the module consistently improves various Transformers. Overall, it achieves an average promotion of 18.2% on PatchTST, 12.5% on TimeXer, and 12.4% on Autoformer. Unlike the mainstream approach of using an encoder-only architecture for patch-level modeling, Autoformer is a point-wise modeling framework based on an encoder-decoder architecture, which further highlights the effectiveness and versatility of the proposed MMoE module.

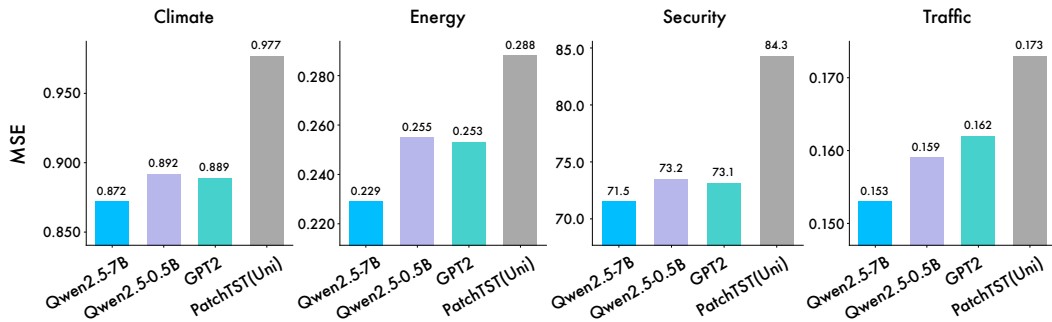

Figure 5: The multimodal time series forecasting results of replacing the LLM backbone. The time series backbone is fixed as PatchTST. The result is the average performance across four dataset-specific prediction horizons. For detailed results, please refer to the Appendix G.3.

Table 2: Performance improvement achieved by applying the MMoE module. Each result represents the average across the four prediction lengths, with "Promotion" indicating the percentage enhancement relative to the original model. Complete results are provided in the Appendix G.1.

| Datasets | | Agriculture | | Climate | | Energy | | Security | | SocialGood | | Traffic | |
|---|---|---|---|---|---|---|---|---|---|---|---|---|---|
| Metric | | MSE | MAE | MSE | MAE | MSE | MAE | MSE | MAE | MSE | MAE | MSE | MAE |
| TimeXer | Original | 0.230 | 0.306 | 0.950 | 0.767 | 0.247 | 0.361 | 78.924 | 4.305 | 1.302 | 0.514 | 0.176 | 0.220 |
| | **+MMoE** | **0.203** | **0.304** | **0.878** | **0.738** | **0.226** | **0.342** | **71.635** | **3.929** | **0.962** | **0.512** | **0.155** | **0.219** |
| | Promotion | 11.7% | 0.7% | 7.6% | 3.8% | 8.5% | 5.3% | 9.2% | 8.7% | 26.1% | 0.4% | 11.9% | 0.5% |
| PatchTST | Original | 0.242 | 0.312 | 0.977 | 0.779 | 0.288 | 0.392 | 84.305 | 4.560 | 1.293 | 0.523 | 0.173 | 0.219 |
| | **+MMoE** | **0.193** | **0.299** | **0.872** | **0.737** | **0.229** | **0.344** | **71.519** | **3.896** | **0.886** | **0.449** | **0.154** | **0.218** |
| | Promotion | 20.2% | 4.2% | 10.7% | 5.4% | 20.5% | 12.2% | 15.2% | 14.6% | 31.5% | 14.1% | 11.0% | 0.5% |
| Autoformer | Original | 0.282 | 0.350 | 0.943 | 0.763 | 0.306 | 0.418 | 83.722 | 4.379 | 1.131 | 0.559 | 0.165 | 0.230 |
| | **+MMoE** | **0.233** | **0.334** | **0.878** | **0.740** | **0.251** | **0.386** | **75.867** | **4.193** | **0.989** | **0.557** | **0.148** | **0.222** |
| | Promotion | 17.4% | 4.6% | 6.9% | 3.0% | 18.0% | 7.6% | 9.4% | 4.2% | 12.6% | 0.4% | 10.3% | 3.5% |

*(ii) Varying LLM backbone.* TiMi leverages reasoning ability of large language models (LLMs) to process complex contextual information from textual data and infer future variations, which subsequently guide time series predictions. To validate the utilization of LLMs, we fix PatchTST as time series backbone and three widely used LLMs, including Qwen2.5-7B (Qwen et al., 2025), Qwen2.5-0.5B (Qwen et al., 2025) and GPT-2 (Radford et al., 2019). As illustrated in Figure 5, benefing from the advanced reasoning capabilities, the Qwen2.5-7B model achieve the best results. Meanwhile, Qwen2.5-0.5B and GPT-2, as smaller-scale LLMs, deliver comparable performance on average and consistently outperform the unimodal baseline. These findings demonstrate that incorporating textual knowledge significantly enhances time-series prediction.

## 4.3 MODEL ANALYSIS

**Ablation Study** To evaluate the performance of each component in the TiMi architecture, we conduct three detailed ablation studies: *(i)* removing components (denoted as *w/o*), *(ii)* replacing components (denoted as *Replace*), and *(iii)* initializing the LLM randomly instead of using the pretrained version (denoted as *Rand. LLM*). As shown in Table 3, the combination of the two modules, TMoE and SMoE, generally yields superior results. In the component replacement experiments, substituting the two carefully designed modules with vanilla MoE (Fedus et al., 2022) leads to suboptimal results. While retaining the SMoE module, replacing the text module with a cross-attention module also yields inferior results compared with TiMi. The LLM randomly initialization experiment verifies the significance of the knowledge that LLMs acquire through pre-training.

Table 3: Ablations results on TiMi. The experiments includes the removal of specific components, the replacement of components, and the random initialization of the LLM. Each result reports the average performance across the dataset's four prediction lengths.

| Design | Textual | Temporal | Agriculture | | Energy | | Health(US) | | SocialGood | | Traffic | |
|---|---|---|---|---|---|---|---|---|---|---|---|---|
| | | | MSE | MAE | MSE | MAE | MSE | MAE | MSE | MAE | MSE | MAE |
| **TiMi** | **TMoE** | **SMoE** | **0.193** | **0.299** | **0.229** | **0.344** | **1.194** | **0.745** | **0.886** | **0.449** | **0.154** | **0.218** |
| w/o | w/o TMoE | SMoE | 0.204 | 0.301 | 0.231 | 0.346 | 1.267 | 0.759 | 0.988 | 0.463 | **0.154** | 0.226 |
| | | w/o | 0.200 | 0.300 | 0.251 | 0.366 | 1.255 | 0.764 | 0.892 | 0.473 | 0.167 | 0.214 |
| Replace | MoE | MoE | 0.231 | 0.311 | 0.264 | 0.373 | 1.268 | 0.763 | 0.958 | 0.471 | 0.162 | 0.222 |
| | Cross | SMoE | 0.227 | 0.311 | 0.283 | 0.382 | 1.308 | 0.780 | 0.951 | 0.436 | 0.159 | 0.226 |
| Rand. LLM | TMoE | SMoE | 0.212 | 0.307 | 0.262 | 0.371 | 1.293 | 0.783 | 1.022 | 0.459 | 0.160 | 0.230 |

**Series-aware Analysis** To assess the interpretability of the proposed Series-Aware Mixture of Experts (SMoE) module, we design an analytical model comprising four experts and adopt a top-1 routing strategy so that only one expert is selected for each input. During evaluation, we record the expert index activated for each test sample and visualize the corresponding routed time series. The visualization reveals that time series routed to the same expert tend to exhibit simi-

lar global trends. Motivated by this observation, we employ the Mann-Kendall (MK) trend test (Mann, 1945; Kendall, 1948), a widely recognized non-parametric statistical method for identifying monotonic trends. The MK test yields a normalized statistic $\mathcal{Z}$, which the calculation process detailed in the Appendix E. Specifically, $\mathcal{Z} > 0$, $\mathcal{Z} < 0$, and $\mathcal{Z} = 0$ indicate an increase trend, a decreasing trend, and no significant trend, respectively. The MK test is particularly effective for evaluating the global trend of a time series, as it provides a robust measure of the overall direction of change without requiring assumptions about linearity or specific data distributions.

To investigate the relationship between global trends and expert selection in the SMoE module, we split the time series into four trend categories based on their MK-derived Z statistics, including strong upward, weak upward, weak downward, and strong downward. As depicted in Figure 6, the experts selection exhibits a clear trend-dependent pattern that series with similar global trends are consistently routed to the same series-level expert. For example, strongly upward-trending series (red flow) predominantly activate Expert-1, while strongly downward-trending series (blue flow) are routed to Expert-4. This empirical finding provides interpretable mechanistic insights into the design of the SMoE module, highlighting its ability to specialize experts and effectively leverage global temporal trends.

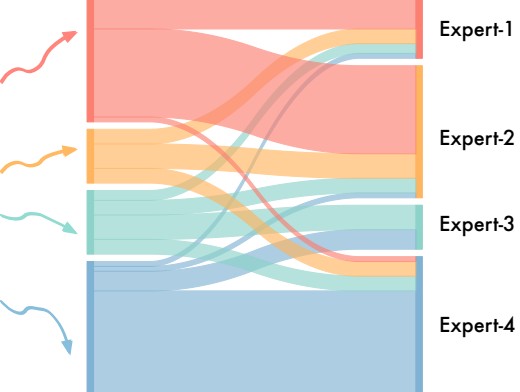

Figure 6: Visualization of the role of Series-aware MoE. Time series are grouped into four trend regimes based on MK test, represented by distinct colors. Series with different trend are consistently routed to different experts.

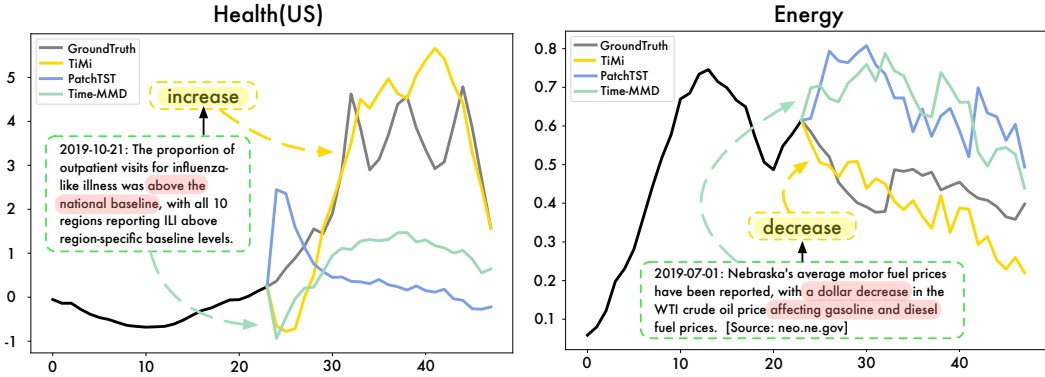

Figure 7: Visualization results on the Health(US) and Energy datasets.

**Case Study of Textual Information** Textual information is one of the factors that has a significant impact on the development direction of numerical data in real systems. In Figure 7, we visualize the multimodal and baseline prediction results, and combine raw text and LLM reasoning results to verify the effect of incorporating textual information into the time series model. It can be seen from the figure that not using text and using the original text cannot better capture trend information. Through the global modeling of time series and the extraction of text information, TiMi has better predicted future trends.

## 5 CONCLUSION

In this paper, we propose TiMi, a time series multimodal forecasting framework that leverages the causal reasoning capabilities of LLMs and incorporates textual content as exogenous guidance. Unlike previous approaches that rely on vague modality fusion, TiMi exploits extracted causal knowledge from textual content of future trends, enhancing both accuracy and interpretability.

Looking ahead, TiMi can be extended by embedding its core module into foundation models for large-scale pretraining or domain-specific finetuning, opening new opportunities for handling massive text–time series corpora and building more general-purpose multimodal forecasting systems.

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

## A THE USE OF LARGE LANGUAGE MODELS

I confirm that a large language model (LLM) was exclusively utilized to refine the writing of this article. All research concepts, ideas, and analyses were independently developed and conducted by the authors. The LLM's role was strictly limited to enhancing clarity and improving style. All substantive contributions and decisions regarding the content were made solely by the authors.

## B DATASETS DESCRIPTIONS

To evaluate the performance of out proposed TiMi framework, we conduct multimodal time series forecasting experiments on 14 real-word datasets. Among them, there are nine datasets from Time-MMD (Liu et al., 2024a), including:

1. **Agriculture** focuses on the retail broiler composite, providing insights into the poultry market in the United States. The target variable is the "Retail Broiler Composite Price," a key economic indicator for the agriculture and food supply chain. This dataset has a monthly sampling frequency and consists of 496 samples spanning from 1983 to the present. It is univariate, with one primary target variable. The accompanying textual data is sourced from USDA reports such as the "Broiler Market News Report" and "Daily National Broiler Market at a Glance," supplemented with web search results.

2. **Climate** centers on drought levels across the contiguous United States, offering critical insights into environmental conditions and their economic impact. The target variable is the "Drought Level," measured across five dimensions. The data is sampled at a monthly frequency, with 496 samples covering the period from 1983 to the present. Textual data includes reports such as the NOAA "Drought Report" and "National Climate Report," along with web search results.

3. **Economy** examines the U.S. international trade balance, providing critical information about the country's economic health. The target variable is the "International Trade Balance," represented in three dimensions. It has a monthly sampling frequency and includes 423 samples spanning from 1989 to the present. Textual sources include the U.S. Census Bureau's "International Trade in Goods and Services" and "Advance Economic Indicators Report," along with web search results.

4. **Energy** focuses on U.S. gasoline prices, a key indicator of energy market trends and consumer behavior. The target variable is "Gasoline Prices," represented in nine dimensions. The dataset has a weekly sampling frequency and includes 1,479 samples covering the period from 1996 to the present. Textual data sources include the "Weekly Petroleum Status Report" and the "Annual Energy Outlook" from the U.S. Energy Information Administration, complemented by web search results.

5. **Environment** monitors air quality across the United States, with the target variable being the "Air Quality Index (AQI)." It is represented in four dimensions, with a daily sampling frequency and a total of 11,102 samples spanning from 1982 to 2023. Textual data is sourced from press releases by the Department of Environmental Conservation of New York and articles from NBC News, along with web search results.

6. **Health(United States)** focuses on influenza-like illnesses (ILI), providing epidemiological data crucial for public health planning. The target variable is the "Proportion of Influenza Patients," measured in eleven dimensions. The dataset has a weekly sampling frequency, with 1,389 samples spanning from 1997 to the present. Textual data is sourced from the CDC's "Weekly Influenza Surveillance Report" and annual flu season reports, supplemented by search results.

7. **Security** chronicles disaster and emergency grants, providing a lens into security and crisis response dynamics. The primary variable is "Disaster and Emergency Grants," a univariate monthly time series consisting of 297 samples spanning from 1999 to the present. Accompanying textual data, sourced from pertinent reports and web searches, has been processed by large language models to improve its coherence and analytical utility.

8. **Social Good** analyzes unemployment data in the United States, offering critical insights into labor market and social welfare trends. The dataset's core is the "Unemployment Rate," a

univariate monthly indicator comprising 900 samples from 1950 to the present. Associated textual information is drawn from a selection of relevant reports and web search results.

9. **Traffic** examines travel volume in the United States, providing insights into transportation and mobility trends. The target variable is "Travel Volume," a univariate indicator recorded at a monthly frequency with 531 samples spanning from 1980 to the present. Textual data includes the U.S. Department of Transportation's "Weekly Traffic Volume Report" and relevant web search results.

Another seven datasets are from Time-IMM (Chang et al., 2025), including:

1. **GDELT** contains event-based logging from the GDELT Event Database 2.0, spanning from 2021 to 2024. The data collection is triggered by real-world events, such as political crises or protests, resulting in irregular, trigger-based sampling. Numerical time series for eight entities (four countries and two event types) include five quantitative features per event. The textual data is sourced from news articles associated with each event, which are then summarized into five-sentence narratives.

2. **RepoHealth** captures adaptive or reactive sampling of GitHub repositories. The sampling frequency is dynamic, with shorter intervals during high-activity "development bursts" and longer intervals during quieter periods. The numerical time series is a 10-dimensional representation of repository metadata, including issue activity and pull request statistics. Textual data is collected from issue and pull request comments, which are aggregated and summarized to provide a contextual narrative for each sampled interval.

3. **FNSPID** uses an operational window sampling method. Observations are limited to predefined operating hours, with no data collected on weekends, holidays, or outside of business hours, leading to structured temporal gaps. Each entity corresponds to a publicly traded company. The numerical time series consists of daily closing prices and trading statistics. The textual data comes from news headlines and article excerpts related to the company, which are summarized into five-sentence narratives.

4. **ClusterTrace** is an example of resource-aware collection, where telemetry is logged only when compute resources are active. Data are missing not because of failure, but because the monitoring system is deliberately inactive during idle periods, resulting in a naturally irregular time series. Each entity is a distinct physical GPU machine. The numerical time series consists of 11 dimensions, including sensor metrics and concurrent job counts. Textual data is derived from job and task descriptions, which are aggregated and summarized into concise narratives.

5. **ILINet** represents a scenario with missing data / gaps, where data are expected but not recorded due to reporting failures or technical problems. It contains a single entity: the United States. The numerical time series is a weekly count of influenza-like illness (ILI) cases, with missing values preserved to reflect true observational gaps. The textual data consists of weekly CDC public health summaries and provider notes, which provide asynchronous, contextual insights.

6. **CESNET** exhibits scheduling jitter/delay, where data points are recorded at uneven intervals due to system delays, congestion, or throttling. The time series is composed of flow-level metrics, such as byte and packet counts, and exhibits natural jitter caused by internal scheduling constraints. Each entity is a distinct networked device. The textual data consists of independent, asynchronously occurring network device logs that include maintenance messages and system warnings.

7. **EPA-Air** is a prime example of multi-source asynchrony, where different data streams operate with different clocks and sampling rates. Each entity is a U.S. county. The numerical time series for four environmental variables (AQI, Ozone, PM2.5, Temperature) are recorded at their native, sensor-specific frequencies, resulting in asynchronous observations. The textual data is sourced from news articles about the weather in the corresponding counties, which are then summarized to provide contextual background.

## C  IMPLEMENTATION DETAILS

All the experiments are implemented in Pytorch and conducted on a single NVIDIA A100 40GB GPU. The ADAM optimizer is employed with an initial learning rate of $10^{-4}$, and L2 loss is used for model optimization. For the Time-MMD datasets, training is fixed to 10 epochs with early stopping, while for the Time-IMM datasets, training is fixed to 1000 epochs with early stopping to align with the Time-IMM settings. The batch size, the number of time series transformer layers, and the dimensions of series representation are respectively selected from the sets $\{4, 16\}$, $\{1, 2, 3\}$, and $\{64, 128, 512\}$.

To assess the performance of the model, we employ the Mean Squared Error (MSE) and Mean Absolute Error (MAE) metrics, following standard practices established in previous studies. Given that a time series $\mathbf{x}_{1:L} = \{\mathbf{x}_1, \mathbf{x}_2, \ldots, \mathbf{x}_L\}$, these metrics are formally defined as:

$$\text{MSE} = \sum_{i=1}^{L} |\mathbf{x}_i - \widehat{\mathbf{x}}_i|^2, \quad \text{MAE} = \sum_{i=1}^{L} |\mathbf{x}_i - \widehat{\mathbf{x}}_i|.$$

Here, $\widehat{\mathbf{x}}_i$ represents the predicted value at the $i$ time step. We report the standard deviations under three runs with different random seeds in Table 4, which exhibits that the performance of TiMi is stable.

Table 4: Robustness of TiMi performance on the Time-MMD datasets. The results are obtained from three random seeds. The three values of horizon in each row of the table correspond respectively to the three datasets on the right.

| Dataset | Algriculture | | Climate | | Economy | |
|---|---|---|---|---|---|---|
| Horizon | MSE | MAE | MSE | MAE | MSE | MAE |
| 6/6/6 | 0.125±0.002 | 0.244±0.001 | 0.860±0.003 | 0.733±0.002 | 0.011±0.001 | 0.084±0.001 |
| 8/8/8 | 0.174±0.003 | 0.285±0.002 | 0.865±0.001 | 0.734±0.001 | 0.011±0.001 | 0.085±0.001 |
| 10/10/10 | 0.220±0.004 | 0.315±0.003 | 0.879±0.004 | 0.739±0.002 | 0.012±0.001 | 0.088±0.001 |
| 12/12/12 | 0.254±0.010 | 0.350±0.008 | 0.885±0.005 | 0.741±0.005 | 0.012±0.001 | 0.088±0.001 |

| Dataset | Energy | | Environment | | Health(US) | |
|---|---|---|---|---|---|---|
| Horizon | MSE | MAE | MSE | MAE | MSE | MAE |
| 12/48/12 | 0.093±0.006 | 0.220±0.007 | 0.404±0.002 | 0.460±0.001 | 0.938±0.010 | 0.665±0.009 |
| 24/96/24 | 0.196±0.005 | 0.322±0.003 | 0.408±0.002 | 0.464±0.001 | 1.163±0.008 | 0.732±0.005 |
| 36/192/36 | 0.271±0.006 | 0.389±0.004 | 0.408±0.003 | 0.463±0.002 | 1.300±0.009 | 0.787±0.006 |
| 48/336/48 | 0.354±0.004 | 0.446±0.002 | 0.411±0.003 | 0.462±0.001 | 1.374±0.011 | 0.796±0.006 |

| Dataset | Security | | SocialGood | | Traffic | |
|---|---|---|---|---|---|---|
| Horizon | MSE | MAE | MSE | MAE | MSE | MAE |
| 6/6/6 | 67.207±0.002 | 3.766±0.002 | 0.782±0.001 | 0.378±0.001 | 0.135±0.000 | 0.195±0.000 |
| 8/8/8 | 68.888±0.002 | 3.847±0.001 | 0.820±0.002 | 0.408±0.001 | 0.152±0.002 | 0.221±0.001 |
| 10/10/10 | 74.324±0.000 | 3.930±0.000 | 0.925±0.004 | 0.465±0.000 | 0.158±0.002 | 0.221±0.001 |
| 12/12/12 | 75.658±0.001 | 4.040±0.000 | 1.016±0.003 | 0.545±0.000 | 0.170±0.000 | 0.234±0.000 |

## D  PROMPTS OF LARGE LANGUAGE MODELS

In our method, prompt words are used to enable the Large Language Models (LLMs) to infer the information related to the time series prediction in the exogenous text. The prompts follow three core principles to alleviate the common ambiguities and hallucinations in language models:

1. *Domain Anchoring*: Embed dataset-specific context, understand the scenarios and prediction requirements to restrict LLMs to the target time series domain.
2. *Task Constraints*: Define clear, narrow tasks to focus LLMs on extracting key temporal attributes (trend, frequency, noise) critical for forecasting.

3. *Output Standardization*: By predefining options, it is required to output concise and struc­tured content, restricting the space for expression of the language model.

The implementation of the prompts is shown in Figure 8. We construct a universal multi-dimensional time series reasoning prompt framework, which helps LLMs infer future time series changes in multiple dimensions from historical textual information, and thereby guide the text-informed time series prediction.

---

domain_ctx: The target variable is the U.S. retail broiler composite index. Data comes from USDA Economic Research Service, with text inputs including USDA market reports and 'broiler market' related search content. Time series frequency: monthly. Forecast horizon: next 6-12 months.

text_info: 2016-12-19: The export price index for meat, poultry, and other animal products was affected by the 2015 U.S. dollar rise, influencing U.S. export price trends and demand for U.S. exports.  [Source: www.bls.gov];

- - - - - - - - - - - - - - - - - - - - - - - - - - - - - - - - - - - - - - - - - - - - - - - - - - - - - - - - -

[Future Trend Prediction]:

System: You are a trend prediction expert. Your sole responsibility is to select the correct future trend from the given options.

User: Predict the future trend of the target indicator in this domain based on the following information. {domain_ctx} Choose EXACTLY ONE option from: no information about trend, keep steady, decrease, increase, multiple. Information: {text_info}

[Future Frequency Prediction]:

System: You are a frequency prediction specialist. Your ONLY task is to classify future frequency as either 'high frequency' or 'low frequency' based on the information provided.

User: Predict the future frequency of the target phenomenon in this domain based on the following information. {domain_ctx} Choose one option: no information about frequency, keep steady, high frequency, low frequency. Information: {text_info}

[Future Noise Prediction]:

System: You are an expert in noise pattern prediction. Analyze the given information and predict future noise characteristics of the target time series in this domain. Provide a concise answer focused on noise levels, regularity, or intensity changes.

User: Predict the future noise pattern of the target indicator in this domain based on the following information. {domain_ctx} Your answer should briefly describe expected changes in noise (e.g., 'increase in noise level', 'become more regular', 'decrease'). Information: {text_info}

Figure 8: General multi-dimensional time series reasoning prompts. The components include domain-specific prompts, textual information, system prompts, question and option prompts in three time series dimensions (trend, frequency, and noise).

In addition, we present a set of reasoning cases using prompts, as illustrated in Figure 9. The results demonstrate that well-designed prompts can effectively mitigate performance risks caused by noisy or off-topic text.

---

domain_ctx: The target variable is U.S. gasoline price fluctuations. Data provided by EIA, with text inputs including weekly petroleum status reports and 'gasoline price' related searches. Time series frequency: weekly. Forecast horizon: next 12-48 weeks.

origin_text: 2019-07-01: Nebraska's average motor fuel prices have been reported, with a dollar decrease in the WTI crude oil price affecting gasoline and diesel fuel prices. [Source: neo.ne.gov]

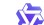 Future Trend Prediction:
Decrease

niosy_text: 2019-e7-01: Nebralska's afvrage motor wrices have been report, with a dollar decrease in thhe WTe bcrudeoil price affect&ng1gasoline anv duiesel fuel prices. wSource: neo.e.gov]

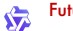 Future Trend Prediction:
Decrease

irrelevant_text: 2019-10-21: The proportion of outpatient visits for influenza-like illness was above the national baseline, with all 10 regions reporting ILI above region-specific baseline levels.

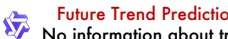 Future Trend Prediction:
No information about trend

Figure 9: An reasoning case on the Energy dataset. Prompts exhibit strong adaptability to variations in text quality, effectively extracting information about key temporal attributes.

## E  SERIES-AWARE ANALYSIS

We employ the Mann-Kendall (MK) trend test (Mann, 1945; Kendall, 1948), a widely well-known statistical method for identifying monotonic trends. The test calculates a normalized statistic $\mathcal{Z}$, which reflects both the strength and direction of the trend. Given a time series represented as $\mathbf{x}_{1:L} = \{\mathbf{x}_1, \mathbf{x}_2, \ldots, \mathbf{x}_L\}$, it can be written as:

$$\mathcal{S} = \sum_{1 \le i < j \le L}^{L-1} \mathrm{sgn}(\mathbf{x}_j - \mathbf{x}_i), \quad \mathcal{Z} = \begin{cases} \frac{\mathcal{S}-1}{\sqrt{\mathrm{Var}(\mathcal{S})}}, & \text{if } \mathcal{S} > 0, \\ 0, & \text{if } \mathcal{S} = 0, \\ \frac{\mathcal{S}+1}{\sqrt{\mathrm{Var}(\mathcal{S})}}, & \text{if } \mathcal{S} < 0. \end{cases}$$

where sgn denotes the sign function, and $\text{Var}(\mathcal{S})$ represents the variance of $\mathcal{S}$ under the null hypothesis of no trend. Specifically, a positive $\mathcal{Z}$ indicates an increasing trend, a negative $\mathcal{Z}$ indicates a decreasing trend, and values of $\mathcal{Z}$ near zero suggest no significant trend.

In addition to the visualization results on Economy in the main text, the visualization results on the other two datasets, Climate and Traffic, are shown in the Figure 10. This verifies the effectiveness of our series-aware routing, which can enhance the expert's focusing ability and help the model better learn the overall trend of the historical time series.

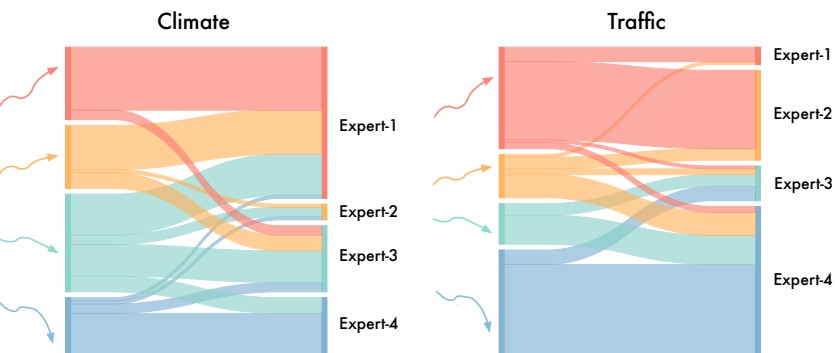

Figure 10: Visualization of Series-aware analysis. The left and right figures are respectively conducted on dataset Climate and dataset Traffic. The strong downward, weak downward, weak upward, and strong upward correspond respectively to $Z \in (-\infty, -1)$, $Z \in [-1, 0)$, $Z \in [0, 1)$, $Z \in [1, +\infty)$.

## F  SHOWCASES

To more clearly demonstrate the differences among various models, we visualize the prediction results on the Energy and Health(US) datasets. We make a fair comparison among three models, TiMi, Time-MMD and PatchTST. Among them, TiMi achieves the best visual prediction effect on the two datasets.

## G  FULL RESULTS

### G.1  FULL RESULTS OF PERFORMANCE IMPROVEMENT

We incorporate the Multimodal Mixture of Experts (MMoE) module into three state-of-the-art time series models: TimeXer (Wang et al., 2024), PatchTST (Nie et al., 2022), and Autoformer (Wu et al., 2021). The results, detailed in the Table 5, demonstrate that MMoE functions as a universal and effective enhancement for Transformer-based time series models. By improving their multimodal modeling capabilities, MMoE offers valuable insights into effectively utilizing textual information to enhance time series forecasting performance.

### G.2  FULL RESULTS OF IRREGULAR FORECASTING

We evaluate TiMi on seven multimodal real-world datasets from Time-IMM (Chang et al., 2025) to assess its predictive performance on challenging irregular time series data. The experiments follow the input and output length configurations of Time-IMM, with the full results summarized in Table 6. Across all seven datasets, TiMi outperforms the baseline model, demonstrating its effectiveness in aggregating irregular textual information and positively influencing time series forecasting. These results provide valuable insights into improving predictions for irregular time series data when combined with textual information.

### G.3  FULL RESULTS OF REPLACING LLM BACKBONE

To assess the influence of the reasoning capability of large language models (LLMs) on the TiMi architecture, the Qwen2.5-7B model used as the LLM backbone in the primary experiment is replaced

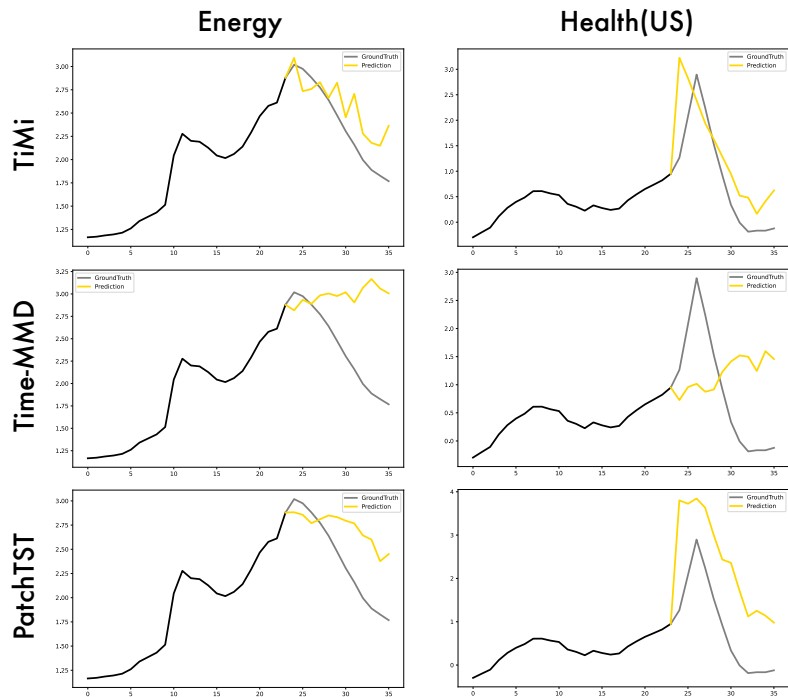

Figure 11: Visualization results on the Energy and Health(US) datasets configured with input-24-predict-12 setting. We make comparisons among the three models, TiMi, Time-MMD and PatchTST.

with the Qwen2.5-0.5B and GPT-2 models. These results are then compared against the single-modal baseline (denoted as "w/o"). The complete experimental outcomes are presented in Table 7. The results demonstrate that more advanced language models generally yield superior performance.

### G.4 FULL RESULTS OF MULTIMODAL FORECASTING

In this section, we fairly compare advanced time series models on nine multimodal time series datasets. In Table 8, we provide the detailed results of our method alongside the latest state-of-the-art multimodal time series forecasting model, $S^2$TS-LLM (Qin et al., 2025), Time-VLM (Zhong et al., 2025) and TimeCMA (Liu et al., 2025), while Table 9 presents the full results from the main experiment.

Table 5: Detailed results of the multimodal time series forecasting performance improvement achieved by applying the Multimodal Mixture of Experts (MMoE) module. *Avg* means the average results from all four prediction lengths.

| Models | | TimeXer (2024) MSE | MAE | **TimeXer +MMoE** MSE | MAE | PatchTST (2022) MSE | MAE | **PatchTST +MMoE** MSE | MAE | Autoformer (2021) MSE | MAE | **Autoformer +MMoE** MSE | MAE |
|---|---|---|---|---|---|---|---|---|---|---|---|---|---|
| Agriculture | 6 | 0.143 | 0.248 | 0.132 | 0.254 | 0.174 | 0.262 | 0.125 | 0.244 | 0.220 | 0.317 | 0.172 | 0.282 |
| | 8 | 0.218 | 0.299 | 0.170 | 0.289 | 0.216 | 0.302 | 0.174 | 0.285 | 0.257 | 0.336 | 0.190 | 0.290 |
| | 10 | 0.240 | 0.316 | 0.228 | 0.321 | 0.262 | 0.327 | 0.220 | 0.315 | 0.304 | 0.358 | 0.265 | 0.367 |
| | 12 | 0.320 | 0.361 | 0.281 | 0.353 | 0.315 | 0.357 | 0.254 | 0.350 | 0.349 | 0.391 | 0.305 | 0.397 |
| | Avg | 0.230 | 0.306 | 0.203 | 0.304 | 0.242 | 0.312 | 0.193 | 0.299 | 0.282 | 0.350 | 0.233 | 0.334 |
| Climate | 6 | 0.918 | 0.750 | 0.861 | 0.730 | 0.938 | 0.760 | 0.860 | 0.733 | 0.975 | 0.777 | 0.868 | 0.737 |
| | 8 | 0.940 | 0.766 | 0.876 | 0.740 | 0.974 | 0.776 | 0.865 | 0.734 | 0.916 | 0.750 | 0.876 | 0.740 |
| | 10 | 0.994 | 0.787 | 0.882 | 0.739 | 1.014 | 0.799 | 0.879 | 0.739 | 0.950 | 0.768 | 0.885 | 0.740 |
| | 12 | 0.950 | 0.765 | 0.894 | 0.744 | 0.982 | 0.779 | 0.885 | 0.741 | 0.931 | 0.757 | 0.884 | 0.742 |
| | Avg | 0.950 | 0.767 | 0.878 | 0.738 | 0.977 | 0.779 | 0.872 | 0.737 | 0.943 | 0.763 | 0.878 | 0.740 |
| Energy | 12 | 0.091 | 0.212 | 0.085 | 0.213 | 0.102 | 0.225 | 0.093 | 0.220 | 0.160 | 0.286 | 0.115 | 0.262 |
| | 24 | 0.184 | 0.321 | 0.184 | 0.315 | 0.278 | 0.394 | 0.196 | 0.322 | 0.270 | 0.404 | 0.213 | 0.358 |
| | 36 | 0.322 | 0.427 | 0.269 | 0.386 | 0.311 | 0.419 | 0.271 | 0.389 | 0.356 | 0.461 | 0.293 | 0.430 |
| | 48 | 0.391 | 0.485 | 0.364 | 0.453 | 0.461 | 0.532 | 0.354 | 0.446 | 0.438 | 0.519 | 0.383 | 0.496 |
| | Avg | 0.247 | 0.361 | 0.226 | 0.342 | 0.288 | 0.392 | 0.229 | 0.344 | 0.306 | 0.418 | 0.251 | 0.386 |
| Security | 6 | 75.615 | 4.162 | 66.922 | 3.741 | 92.333 | 4.799 | 67.207 | 3.766 | 83.241 | 4.291 | 75.085 | 4.164 |
| | 8 | 79.090 | 4.361 | 68.720 | 3.894 | 81.402 | 4.506 | 68.888 | 3.847 | 81.359 | 4.264 | 74.602 | 4.161 |
| | 10 | 80.424 | 4.296 | 74.873 | 4.004 | 80.814 | 4.428 | 74.324 | 3.930 | 80.692 | 4.263 | 76.223 | 4.155 |
| | 12 | 80.568 | 4.399 | 76.098 | 4.079 | 82.669 | 4.506 | 75.658 | 4.040 | 89.597 | 4.697 | 77.557 | 4.293 |
| | Avg | 78.924 | 4.305 | 71.653 | 3.929 | 84.305 | 4.560 | 71.519 | 3.896 | 83.722 | 4.379 | 75.867 | 4.193 |
| SocialGood | 6 | 1.132 | 0.463 | 0.858 | 0.459 | 1.234 | 0.475 | 0.782 | 0.378 | 0.970 | 0.473 | 0.875 | 0.530 |
| | 8 | 1.326 | 0.512 | 0.945 | 0.494 | 1.422 | 0.553 | 0.820 | 0.408 | 0.970 | 0.537 | 0.953 | 0.517 |
| | 10 | 1.290 | 0.519 | 1.014 | 0.542 | 1.271 | 0.546 | 0.925 | 0.465 | 1.232 | 0.558 | 1.045 | 0.546 |
| | 12 | 1.458 | 0.559 | 1.033 | 0.551 | 1.244 | 0.518 | 1.016 | 0.545 | 1.353 | 0.668 | 1.082 | 0.634 |
| | Avg | 1.302 | 0.514 | 0.962 | 0.512 | 1.293 | 0.523 | 0.886 | 0.449 | 1.131 | 0.559 | 0.989 | 0.557 |
| Traffic | 6 | 0.161 | 0.210 | 0.137 | 0.202 | 0.159 | 0.211 | 0.135 | 0.195 | 0.153 | 0.223 | 0.133 | 0.206 |
| | 8 | 0.171 | 0.214 | 0.152 | 0.215 | 0.167 | 0.211 | 0.152 | 0.221 | 0.151 | 0.215 | 0.147 | 0.225 |
| | 10 | 0.171 | 0.213 | 0.161 | 0.222 | 0.174 | 0.216 | 0.158 | 0.221 | 0.169 | 0.244 | 0.152 | 0.222 |
| | 12 | 0.203 | 0.244 | 0.170 | 0.239 | 0.194 | 0.238 | 0.170 | 0.234 | 0.185 | 0.239 | 0.160 | 0.235 |
| | Avg | 0.176 | 0.220 | 0.155 | 0.219 | 0.173 | 0.219 | 0.154 | 0.218 | 0.165 | 0.230 | 0.148 | 0.222 |

Table 6: Full results of irregular multimodal time series forecasting on Time-IMM datasets. The complete MSE and MAE indicators are presented.

| Datasets | GDELT MSE | MAE | RepoHealth MSE | MAE | FNSPID MSE | MAE | ClusterTrace MSE | MAE | ILINet MSE | MAE | CESNET MSE | MAE | EPA-Air MSE | MAE |
|---|---|---|---|---|---|---|---|---|---|---|---|---|---|---|
| **TiMi (Ours)** | **1.0293** | **0.6889** | **0.5323** | **0.4161** | **0.1269** | **0.2217** | **0.6890** | **0.6756** | **0.9620** | **0.6651** | **0.8957** | **0.7271** | **0.5987** | **0.5792** |
| IMM-TSF (2025) | 1.0511 | 0.6944 | 0.5520 | 0.4180 | 0.1277 | 0.2212 | 1.0370 | 0.8389 | 1.1727 | 0.7373 | 1.1242 | 0.8192 | 0.6204 | 0.5992 |
| Time-MMD (2024a) | 1.2352 | 0.7731 | 0.5786 | 0.4531 | 0.1413 | 0.2451 | 0.8304 | 0.7447 | 1.0404 | 0.6753 | 0.9706 | 0.7614 | 0.6506 | 0.5975 |
| PatchTST (2022) | 1.0670 | 0.7051 | 0.5727 | 0.4185 | 0.1301 | 0.2240 | 2.0366 | 0.9896 | 1.1606 | 0.7532 | 1.3177 | 0.8797 | 0.6196 | 0.5947 |

Table 7: Full results of replacing LLM backbone for time series forecasting. Each result is the average of the four predicted lengths of this dataset. *w/o* represents the baseline result of the single-modality.

| Datasets | Agriculture MSE | MAE | Climate MSE | MAE | Economy MSE | MAE | Energy MSE | MAE | Health(US) MSE | MAE | Security MSE | MAE | SocialGood MSE | MAE | Traffic MSE | MAE |
|---|---|---|---|---|---|---|---|---|---|---|---|---|---|---|---|---|
| **Qwen2.5-7B** | **0.193** | **0.299** | **0.872** | **0.737** | **0.012** | **0.086** | **0.229** | **0.344** | **1.194** | **0.745** | **71.519** | **3.896** | **0.886** | **0.449** | **0.153** | 0.220 |
| Qwen2.5-0.5B | 0.223 | 0.309 | 0.892 | 0.774 | 0.012 | 0.089 | 0.255 | 0.364 | 1.270 | 0.767 | 73.520 | 4.026 | 1.002 | 0.473 | 0.159 | **0.217** |
| GPT2 | 0.241 | 0.318 | 0.889 | 0.743 | 0.013 | 0.091 | 0.253 | 0.367 | 1.270 | 0.770 | 73.127 | 3.970 | 0.939 | 0.462 | 0.162 | 0.218 |
| w/o LLM | 0.242 | 0.312 | 0.977 | 0.779 | 0.013 | 0.093 | 0.288 | 0.392 | 1.252 | 0.759 | 84.305 | 4.560 | 1.293 | 0.523 | 0.173 | 0.219 |

Table 8: Full results compared with the latest multimodal time series forecasting models with the best in **bold**. For monthly-sampled datasets such as Agriculture, Climate, Economy, Security, SocialGood, and Traffic, the prediction horizons are $H \in \{6, 8, 10, 12\}$. For weekly-sampled datasets such as Energy and Health(US), $H \in \{12, 24, 36, 48\}$. For the daily-sampled Environment dataset, $H \in \{48, 96, 192, 336\}$. *Avg* means the average results from all four prediction lengths.

| Datasets | | Agriculture | | Climate | | Economy | | Energy | | Environment | | Health(US) | | Security | | SocialGood | | Traffic | |
|---|---|---|---|---|---|---|---|---|---|---|---|---|---|---|---|---|---|---|---|
| Metric | | MSE | MAE | MSE | MAE | MSE | MAE | MSE | MAE | MSE | MAE | MSE | MAE | MSE | MAE | MSE | MAE | MSE | MAE |
| **TiMi** **(Ours)** | 6/12/48 | **0.125** | **0.244** | **0.860** | **0.733** | **0.011** | **0.084** | **0.093** | **0.220** | **0.404** | **0.460** | **0.938** | **0.665** | 67.207 | 3.766 | **0.782** | **0.378** | **0.135** | **0.195** |
| | 8/24/96 | **0.174** | **0.285** | **0.865** | **0.734** | **0.011** | **0.085** | **0.196** | **0.322** | **0.408** | **0.464** | **1.163** | **0.732** | 68.888 | 3.847 | **0.820** | **0.408** | **0.152** | **0.221** |
| | 10/36/192 | **0.220** | 0.315 | **0.879** | **0.739** | **0.012** | **0.088** | **0.271** | **0.389** | **0.408** | 0.463 | **1.300** | 0.787 | **74.324** | **3.930** | **0.925** | **0.465** | **0.158** | **0.221** |
| | 12/48/336 | **0.254** | 0.350 | **0.885** | 0.741 | **0.012** | **0.088** | **0.354** | **0.446** | **0.411** | **0.462** | 1.374 | **0.796** | **75.658** | **4.040** | **1.016** | **0.545** | **0.170** | 0.234 |
| | Avg | **0.193** | **0.299** | **0.872** | **0.737** | **0.012** | **0.086** | **0.229** | **0.344** | **0.408** | **0.462** | **1.194** | **0.745** | **71.519** | **3.896** | **0.886** | **0.449** | **0.154** | **0.218** |
| S$^2$TS-LLM (2025) | 6/12/48 | 0.173 | 0.281 | 0.869 | 0.737 | 0.021 | 0.119 | 0.108 | 0.238 | 0.407 | **0.460** | 1.095 | 0.714 | 67.191 | 3.788 | 0.837 | 0.471 | 0.172 | 0.244 |
| | 8/24/96 | 0.220 | 0.318 | 0.870 | 0.738 | 0.019 | 0.112 | 0.218 | 0.341 | 0.416 | 0.468 | 1.262 | 0.765 | 70.926 | 4.040 | 1.055 | 0.569 | 0.174 | 0.241 |
| | 10/36/192 | 0.237 | 0.327 | 0.883 | 0.742 | 0.019 | 0.112 | 0.305 | 0.411 | 0.419 | **0.458** | 1.350 | 0.785 | 79.441 | 4.319 | 0.997 | 0.551 | 0.176 | 0.245 |
| | 12/48/336 | 0.286 | 0.360 | 0.890 | **0.739** | 0.020 | 0.115 | 0.389 | 0.473 | 0.424 | 0.467 | 1.376 | 0.802 | 79.983 | 4.371 | 1.149 | 0.647 | 0.188 | 0.242 |
| | Avg | 0.229 | 0.321 | 0.878 | 0.739 | 0.020 | 0.115 | 0.255 | 0.366 | 0.417 | 0.463 | 1.271 | 0.767 | 74.385 | 4.130 | 1.010 | 0.560 | 0.177 | 0.243 |
| Time-VLM (2025) | 6/12/48 | 0.145 | 0.254 | 0.870 | 0.738 | 0.014 | 0.095 | 0.095 | 0.220 | 0.411 | 0.468 | 1.087 | 0.717 | 73.161 | 4.113 | 0.807 | 0.381 | 0.147 | 0.212 |
| | 8/24/96 | 0.188 | **0.285** | 0.887 | 0.745 | 0.015 | 0.101 | 0.215 | 0.344 | 0.426 | 0.473 | 1.275 | 0.759 | 82.647 | 4.554 | 0.950 | 0.505 | 0.162 | 0.223 |
| | 10/36/192 | 0.239 | **0.309** | 0.938 | 0.768 | 0.014 | 0.095 | 0.307 | 0.413 | 0.422 | 0.474 | 1.350 | 0.790 | 79.966 | 4.398 | 1.001 | 0.501 | 0.165 | 0.229 |
| | 12/48/336 | 0.266 | **0.348** | 0.927 | 0.768 | 0.013 | 0.093 | 0.368 | 0.460 | 0.428 | 0.484 | **1.359** | 0.802 | 81.554 | 4.514 | 1.158 | 0.591 | 0.173 | **0.233** |
| | Avg | 0.210 | 0.299 | 0.906 | 0.755 | 0.014 | 0.096 | 0.246 | 0.359 | 0.422 | 0.475 | 1.268 | 0.767 | 79.332 | 4.395 | 0.979 | 0.494 | 0.162 | 0.224 |
| TimeCMA (2025) | 6/12/48 | 0.169 | 0.298 | 0.883 | 0.743 | 0.027 | 0.134 | 0.223 | 0.366 | 0.427 | 0.476 | 1.384 | 0.845 | **66.220** | **3.702** | 0.927 | 0.558 | 0.161 | 0.260 |
| | 8/24/96 | 0.241 | 0.323 | 0.875 | 0.738 | 0.024 | 0.127 | 0.303 | 0.427 | 0.429 | 0.473 | 1.445 | 0.829 | **68.666** | 3.850 | 1.053 | 0.615 | 0.169 | 0.253 |
| | 10/36/192 | 0.283 | 0.362 | 0.896 | 0.746 | 0.023 | 0.125 | 0.372 | 0.475 | 0.422 | 0.463 | 1.438 | 0.831 | 78.213 | 4.208 | 1.191 | 0.676 | 0.168 | 0.257 |
| | 12/48/336 | 0.333 | 0.389 | 0.890 | 0.745 | 0.023 | 0.124 | 0.453 | 0.526 | 0.425 | 0.470 | 1.461 | 0.835 | 84.520 | 4.691 | 1.185 | 0.706 | 0.175 | 0.263 |
| | Avg | 0.256 | 0.343 | 0.886 | 0.743 | 0.024 | 0.127 | 0.338 | 0.448 | 0.426 | 0.470 | 1.432 | 0.835 | 74.405 | 4.113 | 1.089 | 0.639 | 0.168 | 0.258 |

Table 9: Full results of multimodal time series forecasting. There are three fixed input lengths of 8, 24, and 96, each corresponding to a specific set of prediction lengths $\{6, 8, 10, 12\}$, $\{12, 24, 36, 48\}$, and $\{48, 96, 192, 336\}$, respectively, following the setting of Time-MMD (Liu et al., 2024a). *Avg* means the average results from all four prediction lengths.

| Models | | TiMi (Ours) | | Time-MMD (2024a) | | AutoTimes (2024d) | | Time-LLM (2023) | | GPT4TS (2023) | | TimeXer (2024) | | iTransformer (2024c) | | PatchTST (2022) | | TimesNet (2023a) | | DLinear (2023) | | Autoformer (2021) | |
|---|---|---|---|---|---|---|---|---|---|---|---|---|---|---|---|---|---|---|---|---|---|---|---|
| Metric | | MSE | MAE | MSE | MAE | MSE | MAE | MSE | MAE | MSE | MAE | MSE | MAE | MSE | MAE | MSE | MAE | MSE | MAE | MSE | MAE | MSE | MAE |
| Agriculture | 6 | 0.125 | 0.244 | 0.145 | 0.251 | 0.473 | 0.487 | 0.159 | 0.245 | 0.134 | 0.241 | 0.143 | 0.248 | 0.149 | 0.254 | 0.174 | 0.262 | 0.159 | 0.267 | 0.304 | 0.402 | 0.220 | 0.317 |
| | 8 | 0.174 | 0.285 | 0.181 | 0.287 | 0.583 | 0.501 | 0.268 | 0.323 | 0.211 | 0.310 | 0.218 | 0.299 | 0.188 | 0.285 | 0.216 | 0.302 | 0.196 | 0.302 | 0.692 | 0.637 | 0.257 | 0.336 |
| | 10 | 0.220 | 0.315 | 0.254 | 0.318 | 1.294 | 0.833 | 0.272 | 0.323 | 0.241 | 0.317 | 0.240 | 0.316 | 0.250 | 0.322 | 0.262 | 0.327 | 0.239 | 0.331 | 0.484 | 0.482 | 0.304 | 0.358 |
| | 12 | 0.254 | 0.350 | 0.270 | 0.348 | 1.777 | 1.042 | 0.349 | 0.381 | 0.310 | 0.357 | 0.320 | 0.361 | 0.322 | 0.362 | 0.315 | 0.357 | 0.276 | 0.365 | 0.566 | 0.535 | 0.349 | 0.391 |
| | Avg | 0.193 | 0.299 | 0.213 | 0.301 | 1.032 | 0.716 | 0.262 | 0.318 | 0.224 | 0.306 | 0.230 | 0.306 | 0.227 | 0.306 | 0.242 | 0.312 | 0.217 | 0.316 | 0.511 | 0.514 | 0.282 | 0.350 |
| Climate | 6 | 0.860 | 0.733 | 0.956 | 0.783 | 0.904 | 0.741 | 0.906 | 0.747 | 0.904 | 0.747 | 0.918 | 0.750 | 0.928 | 0.758 | 0.938 | 0.760 | 0.937 | 0.773 | 0.907 | 0.749 | 0.975 | 0.777 |
| | 8 | 0.865 | 0.734 | 0.972 | 0.780 | 0.918 | 0.750 | 0.911 | 0.749 | 0.921 | 0.753 | 0.940 | 0.766 | 0.974 | 0.785 | 0.974 | 0.776 | 0.895 | 0.746 | 0.919 | 0.751 | 0.916 | 0.750 |
| | 10 | 0.879 | 0.739 | 0.944 | 0.772 | 1.017 | 0.789 | 0.929 | 0.756 | 0.944 | 0.763 | 0.994 | 0.787 | 0.970 | 0.777 | 1.014 | 0.799 | 0.951 | 0.775 | 0.916 | 0.752 | 0.950 | 0.768 |
| | 12 | 0.885 | 0.741 | 1.000 | 0.796 | 0.937 | 0.754 | 0.913 | 0.749 | 0.942 | 0.763 | 0.950 | 0.765 | 1.006 | 0.795 | 0.982 | 0.779 | 0.936 | 0.776 | 0.915 | 0.750 | 0.931 | 0.757 |
| | Avg | 0.872 | 0.737 | 0.968 | 0.783 | 0.944 | 0.758 | 0.915 | 0.750 | 0.928 | 0.756 | 0.950 | 0.767 | 0.970 | 0.779 | 0.977 | 0.779 | 0.930 | 0.767 | 0.915 | 0.750 | 0.943 | 0.763 |
| Economy | 6 | 0.011 | 0.084 | 0.018 | 0.115 | 0.463 | 0.564 | 0.018 | 0.110 | 0.014 | 0.093 | 0.015 | 0.098 | 0.012 | 0.087 | 0.013 | 0.093 | 0.014 | 0.094 | 0.062 | 0.200 | 0.087 | 0.239 |
| | 8 | 0.011 | 0.085 | 0.023 | 0.123 | 0.551 | 0.651 | 0.018 | 0.109 | 0.016 | 0.103 | 0.012 | 0.089 | 0.012 | 0.085 | 0.013 | 0.093 | 0.013 | 0.094 | 0.152 | 0.337 | 0.059 | 0.189 |
| | 10 | 0.012 | 0.088 | 0.022 | 0.127 | 0.915 | 0.831 | 0.020 | 0.115 | 0.014 | 0.095 | 0.012 | 0.089 | 0.011 | 0.086 | 0.013 | 0.092 | 0.015 | 0.100 | 0.058 | 0.195 | 0.053 | 0.181 |
| | 12 | 0.012 | 0.088 | 0.023 | 0.126 | 1.016 | 0.868 | 0.017 | 0.105 | 0.013 | 0.093 | 0.012 | 0.088 | 0.012 | 0.089 | 0.013 | 0.094 | 0.015 | 0.096 | 0.070 | 0.216 | 0.046 | 0.169 |
| | Avg | 0.012 | 0.086 | 0.022 | 0.123 | 0.736 | 0.729 | 0.018 | 0.110 | 0.014 | 0.096 | 0.013 | 0.091 | 0.012 | 0.087 | 0.013 | 0.093 | 0.014 | 0.096 | 0.086 | 0.237 | 0.061 | 0.194 |
| Energy | 12 | 0.093 | 0.220 | 0.101 | 0.223 | 0.155 | 0.295 | 0.116 | 0.255 | 0.122 | 0.257 | 0.091 | 0.212 | 0.103 | 0.219 | 0.102 | 0.225 | 0.121 | 0.264 | 0.186 | 0.326 | 0.160 | 0.286 |
| | 24 | 0.196 | 0.322 | 0.221 | 0.346 | 0.299 | 0.422 | 0.235 | 0.368 | 0.241 | 0.374 | 0.184 | 0.321 | 0.208 | 0.339 | 0.278 | 0.394 | 0.228 | 0.355 | 0.267 | 0.394 | 0.270 | 0.404 |
| | 36 | 0.271 | 0.389 | 0.305 | 0.418 | 0.405 | 0.486 | 0.311 | 0.415 | 0.328 | 0.442 | 0.322 | 0.427 | 0.302 | 0.409 | 0.311 | 0.419 | 0.329 | 0.429 | 0.344 | 0.446 | 0.356 | 0.461 |
| | 48 | 0.354 | 0.446 | 0.389 | 0.472 | 0.479 | 0.525 | 0.400 | 0.481 | 0.430 | 0.514 | 0.391 | 0.485 | 0.400 | 0.489 | 0.461 | 0.532 | 0.407 | 0.484 | 0.442 | 0.506 | 0.438 | 0.519 |
| | Avg | 0.229 | 0.344 | 0.254 | 0.365 | 0.335 | 0.432 | 0.266 | 0.380 | 0.280 | 0.397 | 0.247 | 0.361 | 0.253 | 0.364 | 0.288 | 0.392 | 0.271 | 0.383 | 0.310 | 0.418 | 0.306 | 0.418 |
| Environment | 48 | 0.404 | 0.460 | 0.485 | 0.524 | 0.444 | 0.477 | 0.408 | 0.464 | 0.415 | 0.468 | 0.403 | 0.460 | 0.474 | 0.469 | 0.408 | 0.465 | 0.409 | 0.460 | 0.408 | 0.476 | 0.444 | 0.477 |
| | 96 | 0.408 | 0.464 | 0.466 | 0.510 | 0.436 | 0.486 | 0.417 | 0.465 | 0.425 | 0.472 | 0.403 | 0.462 | 0.399 | 0.460 | 0.417 | 0.467 | 0.399 | 0.455 | 0.427 | 0.503 | 0.436 | 0.486 |
| | 192 | 0.408 | 0.463 | 0.470 | 0.515 | 0.432 | 0.485 | 0.415 | 0.464 | 0.426 | 0.467 | 0.400 | 0.466 | 0.400 | 0.464 | 0.424 | 0.474 | 0.397 | 0.461 | 0.451 | 0.535 | 0.432 | 0.485 |
| | 336 | 0.411 | 0.462 | 0.463 | 0.510 | 0.453 | 0.511 | 0.418 | 0.464 | 0.424 | 0.468 | 0.397 | 0.464 | 0.402 | 0.468 | 0.420 | 0.468 | 0.397 | 0.458 | 0.437 | 0.516 | 0.453 | 0.511 |
| | Avg | 0.408 | 0.462 | 0.471 | 0.515 | 0.441 | 0.490 | 0.414 | 0.464 | 0.423 | 0.469 | 0.401 | 0.463 | 0.419 | 0.465 | 0.417 | 0.469 | 0.401 | 0.458 | 0.431 | 0.507 | 0.441 | 0.490 |
| Health(US) | 12 | 0.938 | 0.665 | 1.075 | 0.742 | 1.252 | 0.782 | 1.040 | 0.696 | 1.001 | 0.694 | 1.047 | 0.696 | 0.986 | 0.686 | 1.023 | 0.697 | 1.110 | 0.737 | 1.327 | 0.774 | 1.162 | 0.757 |
| | 24 | 1.163 | 0.732 | 1.267 | 0.800 | 1.362 | 0.771 | 1.213 | 0.738 | 1.216 | 0.743 | 1.296 | 0.793 | 1.197 | 0.725 | 1.237 | 0.738 | 1.180 | 0.747 | 1.346 | 0.765 | 1.316 | 0.794 |
| | 36 | 1.300 | 0.787 | 1.465 | 0.852 | 1.453 | 0.815 | 1.326 | 0.779 | 1.324 | 0.785 | 1.340 | 0.805 | 1.341 | 0.783 | 1.318 | 0.779 | 1.466 | 0.811 | 1.428 | 0.800 | 1.373 | 0.827 |
| | 48 | 1.374 | 0.796 | 1.588 | 0.891 | 1.482 | 0.822 | 1.384 | 0.806 | 1.422 | 0.824 | 1.431 | 0.828 | 1.480 | 0.831 | 1.432 | 0.820 | 1.449 | 0.841 | 1.462 | 0.816 | 1.403 | 0.838 |
| | Avg | 1.194 | 0.745 | 1.349 | 0.821 | 1.387 | 0.798 | 1.241 | 0.754 | 1.241 | 0.761 | 1.279 | 0.781 | 1.251 | 0.756 | 1.252 | 0.759 | 1.301 | 0.784 | 1.390 | 0.789 | 1.313 | 0.804 |
| Security | 6 | 67.207 | 3.766 | 73.380 | 4.228 | 94.614 | 5.282 | 70.788 | 3.820 | 84.543 | 4.617 | 75.615 | 4.162 | 77.524 | 4.237 | 92.333 | 4.799 | 78.768 | 4.366 | 70.917 | 3.875 | 83.241 | 4.291 |
| | 8 | 68.888 | 3.847 | 74.950 | 4.247 | 107.984 | 5.897 | 75.063 | 4.010 | 82.471 | 4.463 | 79.090 | 4.361 | 80.435 | 4.488 | 81.402 | 4.506 | 90.346 | 4.868 | 74.243 | 4.054 | 81.359 | 4.264 |
| | 10 | 74.324 | 3.930 | 83.313 | 4.480 | 97.120 | 5.694 | 75.518 | 4.150 | 92.826 | 5.000 | 80.424 | 4.296 | 82.036 | 4.487 | 80.814 | 4.428 | 88.195 | 4.750 | 75.362 | 4.113 | 80.692 | 4.263 |
| | 12 | 75.658 | 4.040 | 81.450 | 4.480 | 97.332 | 5.611 | 76.106 | 4.132 | 84.248 | 4.651 | 80.568 | 4.399 | 78.900 | 4.245 | 82.669 | 4.506 | 97.498 | 5.037 | 76.544 | 4.180 | 89.597 | 4.697 |
| | Avg | 71.519 | 3.896 | 78.273 | 4.358 | 99.263 | 5.621 | 74.369 | 4.028 | 86.022 | 4.683 | 78.924 | 4.305 | 79.724 | 4.365 | 84.305 | 4.560 | 92.013 | 4.885 | 75.383 | 4.116 | 83.722 | 4.379 |
| SocialGood | 6 | 0.782 | 0.378 | 1.044 | 0.443 | 0.926 | 0.535 | 0.896 | 0.431 | 0.874 | 0.419 | 1.132 | 0.463 | 1.143 | 0.461 | 1.234 | 0.475 | 0.899 | 0.405 | 0.968 | 0.579 | 0.970 | 0.473 |
| | 8 | 0.820 | 0.408 | 1.150 | 0.473 | 0.955 | 0.560 | 0.994 | 0.471 | 1.003 | 0.459 | 1.326 | 0.512 | 1.329 | 0.507 | 1.422 | 0.553 | 1.519 | 0.588 | 1.028 | 0.520 | 0.970 | 0.537 |
| | 10 | 0.925 | 0.465 | 1.030 | 0.491 | 1.033 | 0.620 | 1.059 | 0.504 | 1.170 | 0.523 | 1.290 | 0.519 | 1.205 | 0.503 | 1.271 | 0.546 | 1.618 | 0.523 | 1.038 | 0.566 | 1.232 | 0.558 |
| | 12 | 1.016 | 0.545 | 1.187 | 0.582 | 1.055 | 0.663 | 1.068 | 0.520 | 1.070 | 0.534 | 1.458 | 0.559 | 1.454 | 0.588 | 1.244 | 0.518 | 1.462 | 0.611 | 1.137 | 0.629 | 1.353 | 0.668 |
| | Avg | 0.886 | 0.449 | 1.103 | 0.497 | 0.992 | 0.594 | 1.004 | 0.481 | 1.029 | 0.484 | 1.302 | 0.514 | 1.283 | 0.515 | 1.293 | 0.523 | 1.374 | 0.532 | 1.043 | 0.574 | 1.131 | 0.559 |
| Traffic | 6 | 0.135 | 0.195 | 0.233 | 0.271 | 0.172 | 0.270 | 0.161 | 0.219 | 0.179 | 0.238 | 0.161 | 0.210 | 0.161 | 0.209 | 0.159 | 0.211 | 0.159 | 0.208 | 0.171 | 0.273 | 0.153 | 0.223 |
| | 8 | 0.152 | 0.221 | 0.308 | 0.352 | 0.179 | 0.282 | 0.162 | 0.219 | 0.178 | 0.234 | 0.171 | 0.214 | 0.177 | 0.214 | 0.167 | 0.211 | 0.165 | 0.216 | 0.205 | 0.342 | 0.151 | 0.215 |
| | 10 | 0.158 | 0.221 | 0.288 | 0.331 | 0.184 | 0.286 | 0.175 | 0.222 | 0.184 | 0.233 | 0.171 | 0.213 | 0.173 | 0.212 | 0.174 | 0.216 | 0.169 | 0.213 | 0.179 | 0.296 | 0.169 | 0.244 |
| | 12 | 0.170 | 0.234 | 0.281 | 0.314 | 0.213 | 0.315 | 0.200 | 0.251 | 0.227 | 0.269 | 0.203 | 0.244 | 0.206 | 0.239 | 0.194 | 0.238 | 0.177 | 0.223 | 0.203 | 0.315 | 0.185 | 0.239 |
| | Avg | 0.154 | 0.218 | 0.277 | 0.317 | 0.187 | 0.288 | 0.174 | 0.228 | 0.192 | 0.244 | 0.176 | 0.220 | 0.179 | 0.220 | 0.173 | 0.219 | 0.168 | 0.215 | 0.189 | 0.306 | 0.165 | 0.230 |
| 1st Count | | 36 | 28 | 0 | 1 | 0 | 0 | 0 | 1 | 0 | 1 | 6 | 4 | 4 | 4 | 0 | 3 | 4 | 7 | 0 | 0 | 1 | 0 |

