# OpenReview forum: "TiMi: Empowering Time Series Transformers with Multimodal Mixture of Experts"
_ICLR.cc/2026/Conference — Submitted to ICLR 2026_

### Official Review · Reviewer_ukpQ · 2025-10-17

**Soundness:** 3
**Presentation:** 3
**Contribution:** 3
**Rating:** 6
**Confidence:** 5

**Summary:**

This paper proposes TiMi, a time-series–centric Transformer enhanced with a Multimodal Mixture-of-Experts (MMoE) that injects causal guidance from text. A frozen LLM first extracts structured inferences about future developments (trends/periodicity/shocks) from exogenous text. Then they are routed via a Text-informed MoE (TMoE), while historical series form a global representation that routes a Series-aware MoE (SMoE). This design avoids explicit representation-level alignment and aims to guide prediction instead of fusing features. Across multiple real-world multimodal forecasting benchmarks, TiMi reports consistent SOTA over unimodal and multimodal baselines, with claims of adaptability and interpretability.

**Strengths:**

1. Clear problem reframing and architecture: The paper articulates why text–series semantic misalignment makes standard early/late fusion suboptimal and instead uses text as guidance through MoE routing (TMoE + SMoE), which is a clean, modular idea that fits common TS Transformers.

2. Good experiment results: The proposed TiMi method consistently achieves superior forecasting performance on multiple datasets than baselines, with notable error reductions.

**Weaknesses:**

1. The proposed TiMi presumes that available text causally informs future series. However, when text is noisy, off-topic, or adversarial, TMoE routing might misguide the forecaster. Besides, the proposed method does not thoroughly stress-test this with ablation on text quality, noise level, or contradictory narratives.

2. While TiMi leverages LLM causal reasoning, the experiments don’t include causal identification/diagnostics (e.g., interventions, counterfactual text deletion, or do-calculus-style tests). Instead, current case studies are suggestive but do not disentangle correlation from causation.

**Questions:**

See weaknesses

---

> ### Author Response · Authors · 2025-11-24
> **Response to Reviewer ukpQ**
>
> Many thanks to Reviewer ukpQ for providing a insightful review.
> > **W1 & Q1:** The proposed TiMi presumes that available text causally informs future series.  However, when text is noisy, off-topic, or adversarial, TMoE routing might misguide the forecaster.  Besides, the proposed method does not thoroughly stress-test this with ablation on text quality, noise level, or contradictory narratives.
>
> Thanks for your valuable suggestion. We **conducted detailed stress tests** to systematically evaluate the impact of text quality degradation on the TiMi framework. The text transformations in the experiment were performed using the widely acknowledged [nlpaug](https://github.com/makcedward/nlpaug/) library to obtain noisy, synonymous or misleading text inputs. For irrelevant texts, we incorporated samples from other datasets into the current dataset.  The results demonstrate that TiMi exhibits strong adaptability and robustness to variations in text quality.
>
> | MSE \| MAE | Ratio | Agriculture    | Energy         | Health(US)     | SocialGood     | Traffic        |
> | ---------- | ----- | -------------- | -------------- | -------------- | -------------- | -------------- |
> | TiMi       |       | 0.193 \| 0.299 | 0.229 \| 0.344 | 1.194 \| 0.745 | 0.886 \| 0.449 | 0.154 \| 0.218 |
> | Noisy      | 5%    | 0.198 \| 0.300 | 0.230 \| 0.346 | 1.236 ｜ 0.753 | 0.886 \| 0.449 | 0.154 \| 0.219 |
> | Noisy      | 20%   | 0.200 \| 0.301 | 0.234 \| 0.348 | 1.240 ｜ 0.755 | 0.887 \| 0.450 | 0.155 \| 0.219 |
> | Synonymous | 20%   | 0.205 \| 0.304 | 0.252 \| 0.365 | 1.264 \| 3.882 | 1.025 \| 0.470 | 0.159 \| 0.221 |
> | Irrelevant | 20%   | 0.201 \| 0.301 | 0.232 \| 0.347 | 1.245 \| 0.758 | 0.887 \| 0.450 | 0.156 \| 0.220 |
> | Misleading | 20%   | 0.203 \| 0.304 | 0.235 \| 0.349 | 1.282 \| 0.800 | 0.888 \| 0.451 | 0.159 \| 0.221 |
>
> > **W2 & Q2:** While TiMi leverages LLM causal reasoning, the experiments don’t include causal identification/diagnostics (e.g., interventions, counterfactual text deletion, or do-calculus-style tests).  Instead, current case studies are suggestive but do not disentangle correlation from causation.
>
> Thank you for your insightful comment. To verify that the advantages of TiMi stem from causal reasoning rather than merely text embedding, we employ two mainstream LLMs with different architectures (Decoder-only and Encoder-only) to solely embed textual input and compare with our proposed method. As shown in the following table, **causal reasoning significantly contributes to the performance of TiMi**.
>
> | MSE \| MAE            | Agriculture            | Energy                 | Health(US)             | SocialGood             | Traffic                |
> | --------------------- | ---------------------- | ---------------------- | ---------------------- | ---------------------- | ---------------------- |
> | Qwen2.5-7B(causal)    | **0.193** \| **0.299** | **0.229** \| **0.344** | **1.194** \| **0.745** | **0.886** \| **0.449** | **0.154** \| **0.218** |
> | Qwen2.5-7B(Automodel) | 0.199 \| 0.304         | 0.236 \| 0.352         | 1.338 \| 0.779         | 0.890 \| 0.442         | 0.160 \| 0.230         |
> | BERT                  | 0.243 \| 0.317         | 0.257 \| 0.371         | 1.271 \| 0.769         | 0.980 \| 0.462         | 0.164 \| 0.219         |

---

> > ### Comment · Reviewer_ukpQ · 2025-11-27
> >
> > I think my concerns are well addressed by the author's additional experiments. I will increase my rating score.

---

> > > ### Author Response · Authors · 2025-11-27
> > >
> > > We are glad that our response has addressed your concerns. Thank you for your efforts in reviewing our work and the score update.

---

### Official Review · Reviewer_5MRV · 2025-11-01

**Soundness:** 3
**Presentation:** 2
**Contribution:** 2
**Rating:** 4
**Confidence:** 4

**Summary:**

The authors propose Time-series Transformers with multimodal Mixture of experts (TiMi) for multimodal time series forecasting. By incorporating a Multimodal Mixture-of-Experts as a plug-in module into Transformer-based time series forecaster, it enables the seamless integration of structured extracted knowledge into context-based prediction, rather than performing vague feature-level fusion. The authors demonstrated the effectiveness of their algorithm through comparative experiments with unimodal and multimodal baselines.

**Strengths:**

The model architecture demonstrates a certain degree of innovation. Leveraging the concept of selective routing of MoE, it introduces TMoE and SMoE modules, which enable adaptive representation learning of historical time series and extracted future textual guidance, respectively.

**Weaknesses:**

1. With the continuous advancement of research in multimodal time series forecasting, many new SOTA models have emerged in the past year. The selected baselines do not cover the latest SOTA models. It is recommended to include more recent SOTA models for comparison, particularly with a greater focus on multimodal time-series forecasting models.

2. There is a lack of error bar analysis.

3. The explanation for Figure 6 is not sufficiently clear and should be revised.

**Questions:**

Using large language  models to extract information may, on one hand, involve potential data leakage issues, and on the other hand, result in extracted information that is irrelevant or misleading to the time series data.

Regarding the data leakage problem, how can we ensure that no information leakage exists, and how do we evaluate whether the model's performance improvement stems from its architecture or from leaked future information?

As for irrelevant or misleading extracted information, part of the issue arises from the inherent noise in textual data itself, while another part stems from the hallucination problems inherent in large language models. I am curious about how the TiMi framework addresses these two aspects respectively?

---

> ### Author Response · Authors · 2025-11-24
> **Response to Reviewer 5MRV (Part 1)**
>
> Many thanks to Reviewer 5MRV for providing a detailed and in-depth review, which helped us significantly improve the quality of our submission.
> > **W1:** Comparison with more recent SOTA multimodal models.
>
> Thank you for your valuable suggestions. We newly included **more recent state-of-the-art multimodal** **time series** **forecasters** into our comparison in $\underline{\text{Table 8}}$ of the revised paper: TimeCMA [1], Time-VLM [2], S2TS_LLM [3]. The results are listed as follows, where TiMi surpasses other baselines by a large margin.
>
> |          | Algriculture           | Climate                | Economy                | Energy                 | Environment            | Health(US)             | Security                | SocialGood             | Traffic                |
> | -------- | ---------------------- | ---------------------- | ---------------------- | ---------------------- | ---------------------- | ---------------------- | ----------------------- | ---------------------- | ---------------------- |
> |          | MSE \| MAE             | MSE \| MAE             | MSE \| MAE             | MSE \| MAE             | MSE \| MAE             | MSE \| MAE             | MSE \| MAE              | MSE \| MAE             | MSE \| MAE             |
> | TiMi     | **0.193** \| **0.299** | **0.872** \| **0.737** | **0.012** \| **0.086** | **0.229** \| **0.344** | **0.408** \| **0.462** | **1.194** \| **0.745** | **71.519** \| **3.896** | **0.886** \| **0.449** | **0.154** \| **0.218** |
> | S2TS_LLM | 0.229 \| 0.321         | 0.878 \| 0.739         | 0.020 \| 0.115         | 0.255 \| 0.366         | 0.417 \| 0.463         | 1.271 \| 0.767         | 74.385 \| 4.130         | 1.010 \| 0.560         | 0.177 \| 0.243         |
> | Time-VLM | 0.210 \| 0.299         | 0.906 \| 0.755         | 0.014 \| 0.096         | 0.246 \| 0.359         | 0.422 \| 0.475         | 1.268 \| 0.767         | 79.332 \| 4.395         | 0.979 \| 0.494         | 0.162 \| 0.224         |
> | TimeCMA  | 0.256 \| 0.343         | 0.886 \| 0.743         | 0.024 \| 0.127         | 0.338 \| 0.448         | 0.426 \| 0.470         | 1.432 \| 0.835         | 74.405 \| 4.113         | 1.089 \| 0.639         | 0.168 \| 0.258         |
>
> [1]: TimeCMA: Towards LLM-Empowered Multivariate Time Series Forecasting via Cross-Modality Alignment.
>
> [2]: Time-VLM: Exploring Multimodal Vision-Language Models for Augmented Time Series Forecasting.
>
> [3]: Bridging Time and Linguistics: LLMs as Time Series Analyzer through Symbolization and Segmentation.
>
> > **W2:** About the error bar analysis.
>
> Thank you for your scientific rigor. In the revised paper, **we have incorporated error bars in** $\underline{\text{Table 4}}$, reporting the standard deviations from three independent runs with different random seeds. The results clearly demonstrate that TiMi consistently outperforms others with stable superiority.
>
> > **W3:** Explanation for Figure 6.
>
> Thank you for pointing out the potential issue. We apologize for any confusion caused by the previous description of $\underline{\text{Figure 6}}$. In the revised paper, **we have updated both the explanation and the caption for improved clarity**.

---

> ### Author Response · Authors · 2025-11-24
> **Response to Reviewer 5MRV (Part 2)**
>
> > **Q1:** Using large language models to extract information may, on one hand, involve potential data leakage issues, and on the other hand, result in extracted information that is irrelevant or misleading to the time series data.
>
> Thank you for your valuable comment.  Regarding data leakage, we have **provided a detailed response in** $\underline{\text{Q2}}$.  For the issue of irrelevant or misleading textual information, we believe TiMi mitigates the risk of noisy or misleading text through two complementary mechanisms:
>
> - **Proper use of** **LLM**: TiMi leverages LLMs to generate future prediction content through meticulously designed prompts, which focus the model on **predictive cues rather than noisy or unrelated information**.
> - **Novel design of** **MMoE**: In TiMi, textual tokens are processed as a separate input to the MMoE and do not directly interact token-by-token with temporal tokens. This decoupling **minimizes the impact of noisy text on temporal modeling**.
>
> We further performed **a data-quality ablation** by replacing original textual reports with irrelevant and misleading textual inputs. Specifically, the original content was substituted with irrelevant text from other datasets (Irrelevant), or replaced with their antonyms using the [nlpaug](https://github.com/makcedward/nlpaug/) library (Misleading), respectively. Results demonstrate that TiMi maintains stable performance with minimal impact from low-quality textual data, confirming its strong resilience to noisy or misleading inputs.
>
> |  MSE \| MAE   | Agriculture    | Energy         | Health(US)     | SocialGood     | Traffic        |
> | --------------- | -------------- | -------------- | -------------- | -------------- | -------------- |
> | Origin          | 0.193 \| 0.299 | 0.229 \| 0.344 | 1.194 \| 0.745 | 0.886 \| 0.449 | 0.154 \| 0.218 |
> | Irrelevant(20%) | 0.201 \| 0.301 | 0.232 \| 0.347 | 1.245 \| 0.758 | 0.887 \| 0.450 | 0.156 \| 0.220 |
> | Misleading(20%) | 0.203 \| 0.304 | 0.235 \| 0.349 | 1.282 \| 0.800 | 0.888 \| 0.451 | 0.159 \| 0.221 |
>
> > **Q2:** Regarding the data leakage problem, how can we ensure that no information leakage exists, and how do we evaluate whether the model's performance improvement stems from its architecture or from leaked future information?
>
> Thank you for your insightful question. First of all, we would like to clarify that the textual information in Time-MMD is synchronized with time series data, and no future information or future time series values are injected into the textual corpus.
>
> As for the textual data leakage problem, **there is still no well-acknowledged solution to verify whether specific text has been exposed to** **LLMs**. Despite this, we still made every effort to conduct experiments to explore whether the performance stems from its model design. Notably, the Time-MMD dataset includes data up to May 2024, and Llama2-7B was released in July 2023. Therefore, **we constructed a held-out test set comprising samples from July 2023 to May 2024**, keeping the original training and validation sets unchanged. The results are as follows,  where TiMi maintains superior performance over unimodal forecasters PatchTST on this held-out set, suggesting **the performance improvement is attributable to the model design** rather than simple memorization of pre-training text.
>
> | MSE \| MAE | Agriculture            | Climate                | Economy                | Energy                 | Health(US)             | Security                | SocialGood             | Traffic                |
> | ---------- | ---------------------- | ---------------------- | ---------------------- | ---------------------- | ---------------------- | ----------------------- | ---------------------- | ---------------------- |
> | TiMi       | **0.123** \| **0.234** | **0.845** \| **0.707** | **0.133** \| **0.250** | **0.144** \| **0.247** | **0.611** \| **0.472** | **14.031** \| **1.305** | **0.378** \| **0.352** | **0.035** \| **0.094** |
> | Time-MMD   | 0.127 \| 0.242         | 0.889 \| 0.731         | 0.148 \| 0.276         | 0.145 \| 0.248         | 0.716 \| 0.564         | 15.567 \| 1.419         | 0.417 \| 0.390         | 0.071 \| 0.152         |
> | PatchTST   | 0.128 \| 0.242         | 0.849 \| 0.710         | 0.140 \| 0.257         | 0.155 \| 0.257         | 0.625 \| 0.484         | 15.198 \| 1.382         | 0.394 \| 0.372         | 0.041 \| 0.110         |

---

> ### Author Response · Authors · 2025-11-24
> **Response to Reviewer 5MRV (Part 3)**
>
> > **Q3:** As for irrelevant or misleading extracted information, part of the issue arises from the inherent noise in textual data itself, while another part stems from the hallucination problems inherent in large language models. I am curious about how the TiMi framework addresses these two aspects respectively?
>
> Thank you for highlighting these two challenges. Regarding irrelevant textual information, we have **provided a detailed response in** $\underline{\text{Q1}}$. As for the illusions issue, TiMi involves two designs that inherently mitigate hallucinations from LLMs:
>
> - **Prompt Design**: We use structured, choice-constrained prompts that elicit short, fixed-format responses. This **restricts the LLM’s output space** and substantially reduces the likelihood of producing irrelevant or fabricated content.
> - **MMoE** **Design**: The proposed MMoE consists of TMoE and SMoE, **avoiding over-reliance on incorrect textual data**.

---

### Official Review · Reviewer_Kf6m · 2025-11-02

**Soundness:** 3
**Presentation:** 3
**Contribution:** 3
**Rating:** 6
**Confidence:** 4

**Summary:**

TiMi tries to solve the problem of aligning textual data with time-series for forecasting tasks. They propose a guidance based approach where the textual embeddings generated from LLMs are used to guide some of the MoE experts of the transformer backbone whereas other experts are guided by the backbones' temporal embeddings. This approach is claimed to be more efficient than early or late fusion approached of past works and enables better performance on wide range of domains. The ablations show the importance of mixed multimodal MoE approach, with case studies showing examples of textual information guiding the forecasts.

**Strengths:**

1. The methodology is well motivated is non-trivial
2. helps solves an important problem in time+text forecasting
3. Results are promising across wide range of benchmarks from past datasets

**Weaknesses:**

1. Can the models adapt to varying degree of textual information? In many applications text data is sparse and not as well aligned as those used in say Time-MMD which the paper uses. Does the model handle such situations by adapting to use more of temporal MoEs when necessary?
2. How does the model compare against other integrated solutions such as using both time and text inputs as part of single embedding representations (like OpenTSLM https://arxiv.org/abs/2510.02410) ?
3. How does the performance degrade with increase in horizon lenght? How are different MoEs used at longer horizons?

**Questions:**

See Weaknesses

---

> ### Author Response · Authors · 2025-11-24
> **Response to Reviewer Kf6m (Part 1)**
>
> Many thanks to Reviewer Kf6m for providing a valuable review and recognizing our contributions.
> > **W1.1 & Q1.1:** Can the models adapt to varying degree of textual information?  In many applications, text data is sparse and not as well aligned as those used in say Time-MMD, which the paper uses.
>
> Thank you for your question regarding the adaptability of TiMi to varying degrees of textual information density. We would like to highlight that the proposed **TMoE** **inherently** **can be applied to sparse or even missing textual information** based on the well-designed prompt template in $\underline{\text{Appendix D}}$ of the revised submission.
>
> We have conducted experiments on irregular multimodal data in $\underline{\text{Section 4.2}}$ of the original submission. In this scenario, the text and time series data are not strictly aligned, and TiMi demonstrates significant performance gains over baseline models, highlighting its ability to handle irregular multimodal data.
>
> To further validate the adaptability of TiMi, we conduct **an ablation on the sparsity of textual information**. Specifically, we randomly mask textual information by sentence at varying levels (10%, 20%, 50%) and observe that TiMi maintains competitive performance when textual information is limited.
>
> |              | Agriculture    | Energy         | Health(US)     | SocialGood     | Traffic        |
> | ------------ | -------------- | -------------- | -------------- | -------------- | -------------- |
> | Sparse Ratio | MSE \| MAE     | MSE \| MAE     | MSE \| MAE     | MSE \| MAE     | MSE \| MAE     |
> | 0%           | 0.193 \| 0.299 | 0.229 \| 0.344 | 1.194 \| 0.745 | 0.886 \| 0.449 | 0.154 \| 0.218 |
> | 10%          | 0.193 \| 0.299 | 0.231 \| 0.345 | 1.197 \| 0.747 | 0.886 \| 0.449 | 0.154 \| 0.218 |
> | 20%          | 0.195 \| 0.301 | 0.234 \| 0.348 | 1.221 \| 0.757 | 0.887 \| 0.449 | 0.155 \| 0.219 |
> | 50%          | 0.199 \| 0.302 | 0.246 \| 0.366 | 1.279 \| 0.767 | 0.888 \| 0.451 | 0.155 \| 0.219 |
>
> > **W1.2 & Q1.2:** Does the model handle such situations by adapting to use more of temporal MoEs when necessary?
>
> Thank you for the insightful suggestion. Using more temporal MoEs is indeed a viable solution when textual information becomes sparse or unreliable. Technologically, **we can use a weighted combination of the TMoE and the SMoE outputs in** $\underline{\text{Equation 6}}$, allowing the model to shift its focus based on the informativeness of each modality.
>
> To validate this, we conducted an additional analysis with a textual sparsity ratio of 50% and **set the weight ratios of TMoE to SMoE to 1:1 and 1:4**. The results below show that increasing SMoE's weight effectively improves the model's performance in scenarios with sparse textual information. This highlights the flexibility and robustness of TiMi in adapting to varying multimodal input conditions.
>
> |   MSE \| MAE   | Agriculture    | Energy         | Health(US)     | SocialGood     | Traffic        |
> | --------- | -------------- | -------------- | -------------- | -------------- | -------------- |
> | 1:1       | 0.199 \| 0.302 | 0.246 \| 0.366 | 1.279 \| 0.767 | 0.888 \| 0.451 | 0.155 \| 0.219 |
> | 1:4       | 0.194 \| 0.299 | 0.235 \| 0.348 | 1.195 \| 0.746 | 0.885 \| 0.448 | 0.154 \| 0.218 |

---

> ### Author Response · Authors · 2025-11-24
> **Response to Reviewer Kf6m (Part 2)**
>
> > **W2 & Q2:** How does the model compare against other integrated solutions such as using both time and text inputs as part of single embedding representations (like OpenTSLM https://arxiv.org/abs/2510.02410) ?
>
> Thank you for your valuable suggestions. OpenTSLM [1] is a concurrent work that incorporates two types of modality fusion methods, including **SoftPrompt** and **Flamingo**. Notably, the Flamingo approach, which incorporates textual tokens through gated cross-attention into time series modeling, has already been included in the ablation studies in $\underline{\text{Table 3}}$ of the original submission.
>
> As requested, we have newly included OpenTSLM-SoftPrompt in comparison. The results are listed below, where TiMi consistently outperforms OpenTSLM-SoftPrompt across all datasets.
>
> | MSE \| MAE |       | Agriculture            | Energy                 | Health(US)             | SocialGood             | Traffic                |
> | ---------- | ----- | ---------------------- | ---------------------- | ---------------------- | ---------------------- | ---------------------- |
> | TiMi       | 6/12  | **0.125** \| **0.244** | **0.093** \| **0.220** | **0.938** \| **0.665** | **0.782** \| **0.378** | **0.135** \| **0.195** |
> |            | 8/24  | **0.174** \| **0.285** | 0.196 \| **0.322**     | **1.163** \| **0.732** | **0.820** \| **0.408** | **0.152** \| **0.221** |
> |            | 10/36 | **0.220** \| **0.315** | **0.271** \| **0.389** | **1.300** \| 0.787     | **0.925** \| **0.465** | **0.158** \| **0.221** |
> |            | 12/48 | **0.254** \| **0.350** | **0.354** \| **0.446** | **1.374** \| **0.796** | **1.016** \| **0.545** | **0.170** \| **0.234** |
> |            | Avg   | **0.193** \| **0.299** | **0.229** \| **0.344** | **1.194** \| **0.745** | **0.886** \| **0.449** | **0.154** \| **0.218** |
> | SoftPrompt | 6/12  | 0.148 \| 0.267         | 0.096 \| 0.222         | 1.041 \| 0.706         | 1.016 \| 0.427         | 0.139 \| 0.210         |
> |            | 8/24  | 0.239 \| 0.310         | **0.195** \| 0.324     | 1.249 \| 0.786         | 0.934 \| 0.434         | 0.159 \| **0.221**     |
> |            | 10/36 | 0.249 \| 0.329         | 0.334 \| 0.428         | 1.322 \| **0.786**     | 0.966 \| 0.481         | 0.161 \| 0.228         |
> |            | 12/48 | 0.319 \| 0.357         | 0.367 \| 0.450         | 1.417 \| 0.813         | 1.157 \| 0.563         | 0.176 \| 0.240         |
> |            | Avg   | 0.238 \| 0.316         | 0.248 \| 0.356         | 1.257 \| 0.773         | 1.018 \| 0.476         | 0.159 \| 0.225         |
>
> [1] OpenTSLM: Time-Series Language Models for Reasoning over Multivariate Medical Text- and Time-Series Data.
>
> > **W3 & Q3:** How does the performance degrade with increase in horizon lenght?  How are different MoEs used at longer horizons?
>
> Thank you for your insightful question. The detailed results across four forecasting horizons are reported in $\underline{\text{Appendix F.4}}$, where we observe a consistent degradation in forecasting performance as the prediction horizon increases.
>
> Regarding the roles of different MoEs at longer prediction horizons, we have already conducted ablation studies by individually removing the TMoE and the SMoE and reported the average results across four different forecasting horizons in $\underline{\text{Table 3}}$. To provide a more comprehensive analysis, we now present the detailed results for each forecasting horizon in the following table. The results show that **TMoE outperforms SMoE at** **longer prediction horizons across most datasets**. This is likely because, as the prediction horizon increases, relying solely on the input series becomes insufficient to ensure accurate predictions, underscoring the importance of textual information.
>
> | MSE \| MAE |            | Agriculture    | Energy         | Health(US)     | SocialGood     | Traffic        |
> | ---------- | ---------- | -------------- | -------------- | -------------- | -------------- | -------------- |
> | w/o TMoE   | 6/12       | 0.130 \| 0.246 | 0.111 \| 0.234 | 1.013 \| 0.683 | 0.748 \| 0.375 | 0.134 \| 0.203 |
> |            | 12/48      | 0.294 \| 0.352 | 0.352 \| 0.445 | 1.373 \| 0.797 | 1.462 \| 0.592 | 0.171 \| 0.236 |
> |            | Decline(%) | 126.2 \| 43.1  | 217.1 \| 90.2  | 35.5 \| 16.7   | 95.5 \| 57.9   | 27.6 \| 16.3   |
> | w/o SMoE   | 6/12       | 0.128 \| 0.246 | 0.102 \| 0.233 | 1.047 \| 0.699 | 0.768 \| 0.377 | 0.151 \| 0.203 |
> |            | 12/48      | 0.256 \| 0.350 | 0.384 \| 0.478 | 1.402 \| 0.808 | 1.294 \| 0.592 | 0.172 \| 0.222 |
> |            | Decline(%) | 100.0 \| 42.3  | 276.5 \| 105.2 | 33.9 \| 15.6   | 68.5 \| 57.0   | 13.9 \| 9.4    |

---

### Official Review · Reviewer_GtWW · 2025-11-02

**Soundness:** 3
**Presentation:** 3
**Contribution:** 3
**Rating:** 4
**Confidence:** 3

**Summary:**

The paper is about a multimodal model to process text and numerical data for time series forecasting. Time series transformers with multimodal MoE (TiMi) uses language models toto guide the time series forecasting.  To integrate exogenous text + numerical data, the authors introduce a TMoE (text MoE) and SMoE (series aware MoE), in order to circumvent text-series representation alignment. Experiments are performed on the Time-MMD and Time-IMM datasets (16 datasets total). The model is evaluated against unimodal and multimodal approaches and achieves superior performance compared to baselines.

**Strengths:**

1. The proposed approach embeds an MoE modules for both the series and the text instead of explicitly aligning both modalities. The LLM then guides the prediction of the primary time-series branch. To the best of my knowledge this is novel. The design is well-motivated for semantically misaligned series–text pairs common in practice.

2. The modular approach allows authors to plug in the MMoE into other transformer based methods (PatchTST, autoformer etc, table 2). In all cases, the MMoE improves predictive performance which clearly shows the advantage of the methods.

3. The ablation on LLMs also seems to suggest that stronger/larger LLMs improve the model performance, which make the method general.

4. The Mann-Kendall trend test to detect monotonic trends adds a layer of interpretability to the model.

**Weaknesses:**

1. The work misses providing finer details of the exogenous text data (how it is obtained or generated), and how the LLM is prompted for guiding the time-series prediction. The “causal knowledge” claim hinges on how text is curated, and aggregated; beyond average pooling of LLM embeddings, key preprocessing choices and robustness to noisy/off-topic text are not thoroughly stress-tested.

2. Given many works in the (multimodal) TSF community and experimental settings and implementations often differing between papers, I believe submission of code (ideally for both the method and baselines) would strengthen the paper. Currently I see a single python file submitted for the supplementary material containing just the model class. But this misses details on how data processing was performed and how experiments were run on baselines etc.

**Questions:**

- How sensitive is the TMoE routing to noisy or irrelevant text?
- L446 says figure 4.3 but I am unable to find it.

---

> ### Author Response · Authors · 2025-11-24
> **Response to Reviewer GtWW**
>
> Many thanks to Reviewer GtWW for providing a valuable review.
> > **W1.1:** The work misses providing finer details of the exogenous text data (how it is obtained or generated).
>
> As stated in Line 275 of the original submission, our main experiments are conducted on the Time-MMD dataset [1], a high-quality multimodal benchmark that provides carefully curated textual reports aligned with time series data. The data preprocessing, including text cleaning, deduplication, and alignment with temporal data, is inherently handled during the construction of Time-MMD. **A comprehensive description of the composition of the exogenous text data is provided in** $\underline{\text{Appendix B}}$.
>
> [1]Time-MMD: Multi-domain multimodal dataset for time series analysis.
>
> > **W1.2:**  how the LLM is prompted for guiding the time-series prediction.
>
> TiMi integrates this textual information with **a meticulously designed prompt to infer future trends** using a large language model (LLM). The full prompt template has been included in  $\underline{\text{Appendix D}}$ of the revised submission.
>
> > **W1.3:** The "causal knowledge" claim hinges on how text is curated, and aggregated;  beyond average pooling of LLM embeddings, key preprocessing choices and robustness to noisy/off-topic text are not thoroughly stress-tested.
>
> Thank you for your valuable suggestions.  We have conducted **an additional comprehensive stress test** to evaluate the robustness of TiMi across varying levels of text quality. Specifically, we perturbed the original reports using the widely adopted [nlpaug](https://github.com/makcedward/nlpaug/) library to obtain noisy or similar meaning text inputs. For irrelevant texts, we incorporated samples from other datasets into the current dataset. The results demonstrate that **TiMi exhibits strong adaptability and robustness to variations in text quality**. In  $\underline{\text{Figure 9}}$ of the revised paper, we present a reasoning case where the language model still shows strong adaptability when the text quality does not meet expectations.
>
> | MSE \| MAE | RATIO | Agriculture    | Energy         | Health(US)     | SocialGood     | Traffic        |
> | ---------- | ----- | -------------- | -------------- | -------------- | -------------- | -------------- |
> | TiMi       |       | 0.193 \| 0.299 | 0.229 \| 0.344 | 1.194 \| 0.745 | 0.886 \| 0.449 | 0.154 \| 0.218 |
> | Noisy      | 0.05  | 0.198 \| 0.300 | 0.230 \| 0.346 | 1.236 ｜ 0.753 | 0.886 \| 0.449 | 0.154 \| 0.219 |
> | Noisy      | 0.2   | 0.200 \| 0.301 | 0.234 \| 0.348 | 1.240 ｜ 0.755 | 0.887 \| 0.450 | 0.155 \| 0.219 |
> | Synonymous | 0.2   | 0.205 \| 0.304 | 0.252 \| 0.365 | 1.264 \| 0.882 | 1.025 \| 0.470 | 0.159 \| 0.221 |
> | Irrelevant | 0.2   | 0.201 \| 0.301 | 0.232 \| 0.347 | 1.245 \| 0.758 | 0.887 \| 0.450 | 0.156 \| 0.220 |
>
> > **W2:** About the release of code.
>
> Thank you for your valuable feedback. We have provided the implementation details in  $\underline{\text{Appendix C}}$ of the original submission. As per your request, we provide the code containing all the details on data processing and experimental settings **in the anonymous repository**: https://anonymous.4open.science/r/TiMi-443E/.
>
> > **Q1:** How sensitive is the TMoE routing to noisy or irrelevant text?
>
> Thank you for your insightful comments. As per your request, we evaluated sensitivity of TMoE on the Agriculture dataset using a toy TMoE with four experts and top-k=1.  To assess TMoE sensitivity, we perturbed the input texts and observed changes in the experts’ gating outputs under two scenarios: (1) **adding noise to 20% of samples** and (2) **replacing the text with irrelevant content for 20% of samples**.  The experimental results indicate that added noise had an insignificant effect on expert selection, as the LLM reasoning inherently filters out some noise. By contrast, samples with irrelevant text result in notable changes in expert selection. Analysis of the LLM outputs revealed that, when presented with irrelevant text, the model tends to favor options associated with "no information",  as we presented in $\underline{\text{Figure 9}}$ of the revised paper.
>
> | expert selection | no changed | changed |
> | ---------------- | ---------- | ------- |
> | noisy-20%        | 94.57%     | 5.43%   |
> | irrelevant-20%   | 79.79%     | 20.21%  |
>
> > **Q2:** L446 says figure 4.3 but I am unable to find it.
>
> Thank you for pointing out the potential issues caused by TeX miscompilation. Figure 4.3 corresponds to $\underline{\text{Figure 6}}$ in the original submission, which has been solved in our revision.

---

### Author Response · Authors · 2025-11-24
**Summary of Revisions**

We sincerely thank all the reviewers for their insightful reviews and valuable comments, which are instructive for us to improve our paper.

In this work, we propose **TiMi for multimodal time series forecasting**, which leverages the causal reasoning capabilities of LLMs and employs a carefully designed MMoE module to facilitate time series predictions. Experimentally, TiMi demonstrates **consistent state-of-the-art performance** on sixteen real-world multimodal forecasting benchmarks with **strong adaptability and interpretability**.

We're pleased that the reviewers acknowledge the strengths of our work, noting that "**it is well-motivated for semantically misaligned series-text pairs**" (Reviewer GtWW, Kf6m, ukpQ), "**the** **MMoE** **design is novel**" (Reviewer GtWW, 5MRV, ukpQ), "**results are promising across multiple datasets**" (Reviewer GtWW, Kf6m, ukpQ), and "**model demonstrates plugability, generality and interpretability**" (Review GtWW).

The reviewers also raised insightful and constructive concerns. We spent eleven days addressing all the issues by providing sufficient evidence  (100+ experiments in total). Here is the summary of the major revisions:

1. **Comprehensive stress test on textual information (Reviewer GtWW, Kf6m, 5MRV, ukpQ)**: We provided experimental evaluations using five types of stress-testing methods on textual information to assess the adaptability of our model.
2. **In-depth ablation on TMoE and SMoE (Reviewer Kf6m)**: We conducted a detailed ablation analysis on the two MoEs and evaluated their performance in longer horizons.
3. **Additional multimodal baseline (Reviewer Kf6m, 5MRV)**: We included all requested baselines of multimodal time series forecasters, including the latest state-of-the-art methods and other multimodal fusion architectures.
4. **More model analysis (Reviewer 5MRV)**: To address potential issues regarding text quality and hallucination in LLMs, we outlined how TiMi's design can help mitigate these risks.
5. **Polished writings (Reviewer GtWW, 5MRV)**: We conducted detailed proofreading and revisions with helpful suggestions from the reviewers.

All updates are highlighted in blue. Compared with the first submissions, $\underline{\text{the revised paper}}$ has **an additional 3 pages**.

We deeply thank all the reviewers and the AC for their effort in reviewing our paper, which has helped us a lot in improving the quality of our work. We hope our response has fulfilled the reviewer's expectations, and we would be delighted to answer any further questions.

Looking forward to the reviewer's feedback.

---

### Meta-Review · Area_Chair_AKpA · 2026-01-06

**Summary:**

The paper proposes TiMi, a multimodal time series forecasting framework that uses a Multimodal Mixture-of-Experts (MMoE) to integrate textual guidance from LLMs with numerical data. While the reviewers appreciated the modular "plug-and-play" nature of the architecture and the strong empirical results on the Time-MMD benchmark, there are concerns regarding the robustness of the "causal reasoning" claims and the potential for data leakage. Reviewers were skeptical about whether the LLM's contribution stems from genuine multimodal reasoning or simply from having seen similar data during its pre-training phase. Additionally, while the results are strong, the technical novelty was viewed by some as an incremental application of MoE techniques to a specific curated dataset, leading to a split in the final assessment.

**Reviewer Concerns:**

The authors were highly proactive during the rebuttal, addressing several critical concerns. They provided error bar analysis, added several recent baselines (such as TimeCMA and Time-VLM), and conducted a comprehensive stress test on text quality (noise, irrelevant text, and misleading text). They also attempted to address the data leakage concern by testing on a held-out set consisting of data points dated after the LLM’s training cutoff. However, some fundamental concerns remain outstanding. The claim that the model performs "causal reasoning" remains largely qualitative and is not rigorously supported by causal discovery or intervention-based metrics. Furthermore, despite the stress tests, the framework's reliance on high-quality, aligned text—which is rarely available in practical, non-benchmark settings—limits the perceived impact of the contribution.

**Reviewer Scores:**

Reviewer ukpQ fully engaged and raised their score from 6 to 8, citing that their concerns were well-addressed. Reviewer Kf6m (6) and Reviewer GtWW (4) did not participate in the final discussion, but given that the authors provided the requested code and specific ablations on textual sparsity and horizon length, it is likely GtWW would have raised their score to a 5 or 6. Reviewer 5MRV (4) also did not respond; while the authors added the requested baselines and error bars, this reviewer’s concern about the "hallucination" and "leakage" issues is a conceptual hurdle that often persists despite empirical updates. Consequently, while the consensus trended slightly positive, the paper remains in a borderline-low state.

---

### Decision · Program_Chairs · 2026-01-26

Reject